# ChemisTRAG: Table-based Retrieval-Augmented Generation for Chemistry Question Answering

## Abstract

Recent work has shown that retrieval-augmented generation (RAG) improves the performance of large language models (LLMs) for question answering on chemistry. However, existing chemistry RAG techniques are mainly based on text. It is challenging for the retriever to align the information about chemical entities between the query and the underlying corpora, especially if the naming and representation formats change. To address this problem, we propose ChemisTRAG, a RAG system in which information about chemical entities and reactions is stored explicitly as tables in the knowledge base (KB). Upon a query, ChemisTRAG first extracts chemical entities from the query and then selects relevant rows from the tabular KB. This way, the alignment processing is simplified and the accuracy is improved regardless of different naming conventions of compounds. To balance accurate answer retrieval for exact matches and robust reasoning for similar matches, we propose an adaptive reasoning process for the LLM: it first generates a reasoning prototype, then adapts the reasoning path to retrieval results, and finally infers the final answer contextualized on the example reasoning path. We have constructed a dataset of more than 38,000 compounds and 23,000 reactions from the recent five years of patents, and generated eight types of question-answering tasks to evaluate our system. Results show that ChemisTRAG consistently outperforms text-based RAG across all eight tasks, particularly in handling diverse chemical representations like SMILES and IUPAC.

## 1 Introduction

Large Language Models (LLMs) offer a promising avenue to accelerate chemical research by answering technical questions and predicting compounds and reactions (Han et al., 2025; Ramos et al., 2025). However, their utility is limited by outdated knowledge, which undermines reliability and hinders practical deployment (Wellawatte et al., 2025). Retrieval-Augmented Generation (RAG) has emerged as a viable and forward-looking approach to enhance LLMs (Fan et al., 2024; Lee et al., 2025). It bypasses the need for retraining by using updated knowledge repositories, allowing LLMs to acquire fresh chemistry knowledge. Nevertheless, general-domain RAG methods struggle with semantic matching in chemistry due to a lack of domain-specific knowledge, leading to difficulties in handling chemical terminology and linking synonymous expressions (Zhong et al., 2025).

To address this core challenge of semantic matching in chemistry, we propose ChemisTRAG, a novel RAG framework built around a tabular paradigm, which comprises a tabular knowledge base (KB), a table-based retriever, and an adaptive reasoner. This approach centers on structuring chemical knowledge in tables rather than unstructured text, creating a unified framework where queries can be accurately matched to knowledge regardless of terminology variations. By consistently applying this tabular structure across knowledge representation, retrieval, and reasoning, ChemisTRAG provides an end-to-end solution designed to enhance both the accuracy and robustness of LLMs in chemistry.

The first component of ChemisTRAG is a tabular KB that organizes chemical knowledge in tables, enabling accurate retrieval through structured representation. To address the limitation of outdated chemical data in prior work (Schneider et al., 2016; Lowe, 2017; Jin et al., 2017), we construct a KB from recent USPTO patents (2020-2025). Our construction pipeline involves using two LLMs

to extract and cross-check reaction information from the patent text, resulting in a reaction table. The records in this table are then used to query external compound databases to validate compound existence and gather additional details, forming a compound table. The resulting KB consists of these two interlinked tables, structuring diverse information about chemical entities and reactions for subsequent retrieval.

A table-based retriever then maps the flexible expressions in natural language queries to the consistent structure of the KB. This component operates by parsing a natural language query into a formal tuple that captures its key elements, such as the entity type (for example, compound or reaction), the target entity, and the query intent. This structured representation enables a schema-aligned search against the tabular KB. The retriever then matches this query tuple against the appropriate table, either compound or reaction, to retrieve a set of relevant entries. This process effectively bypasses the ambiguities of free-text matching by leveraging the consistent schema of the KB, thus providing precise contextual evidence for answering.

An adaptive reasoner then leverages the retrieved evidence to guide the LLM in generating answers. Its design aims to ensure accuracy when the KB contains exact matches, while preserving the LLM's inherent reasoning capabilities for scenarios requiring inference over similar or related information, via a three-stage process. First, the reasoner generates a reasoning prototype, which is a step-by-step plan structured around the query's intent. Second, it grounds this prototype by integrating the specific facts from the retrieved KB entries. Finally, the LLM produces the final answer within this guided context. This approach ensures the output is constrained by the retrieved knowledge yet flexible enough to handle cases where direct answers are not available.

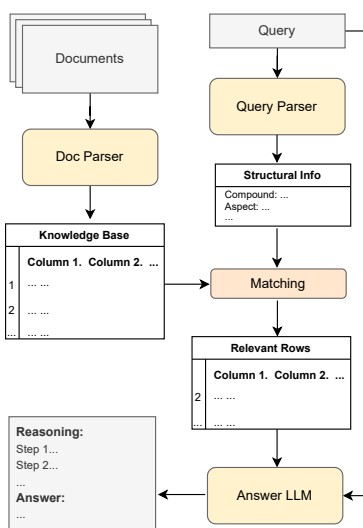

Figure 1: Overview of the Chemis-TRAG framework illustrating the tabular KB, table-based retriever, and adaptive reasoner.

To evaluate our approach and provide in-depth analysis, we construct a benchmark dataset. Following established work (Guo et al., 2023; Zhang et al., 2024; Zhong et al., 2025), we define eight question-answering tasks encompassing fundamental chemistry types such as reaction prediction and compound property inquiry. For each task, we use an LLM to generate slot templates. We then sample records from the KB to populate these templates, creating initial question-answer pairs. To enhance the linguistic diversity and realism, these initial questions are subsequently paraphrased and elaborated by an LLM. This process yields an evaluation set of 4,800 QA pairs, in order to evaluate the effectiveness of ChemisTRAG and measure the impact of retrieval on LLM performance.

We conduct extensive experiments to evaluate our ChemisTRAG for improving LLMs in chemistry question-answering. We compare ChemisTRAG with text-based RAG across eight tasks and two input formats, analyze retrieval recall, test impacts of retrievers and inference strategies, and explore how retrieved entry quantity affects performance. These experiments show ChemisTRAG outperforms text-based RAG significantly. Its table-based design boosts retrieval accuracy across inputs and thus improves LLM performance. Additionally, ChemisTRAG enhances LLMs grounded on both exact-matched and similar-matched knowledge. The consistent superiority fully supports the effectiveness of ChemisTRAG. We summarize our contributions as follows:

- We build an up-to-date structural chemistry knowledge base and a corresponding evaluation dataset for testing RAG systems.
- We propose a tabular RAG paradigm with schema-aligned retrieval and adaptive reasoning to enhance the accuracy of LLM outputs.
- We conduct an in-depth analysis that demonstrates the effectiveness of ChemisTRAG.

## 2 RELATED WORK

We categorize relevant LLM systems for chemistry based on their approaches to knowledge organization, retrieval mechanisms, and reasoning optimization. Table 1 provides a comparative overview.

Table 1: System comparison of knowledge-centric chemical AI systems. **Knowledge Module**: Source (I: internal LLM knowledge, E: external databases) and Form (NL: natural language, SL: structured language). **Retrieval Module**: Query transformation form and support to diverse chemical names. **Reasoning Module**: Whether specialized inference optimization is employed.

| Method | Knowledge Module | | Retrieval Module | | Reasoning Module |
| --- | --- | --- | --- | --- | --- |
| | Source | Form | Query Trans. | Diverse Name | |
| StructChem | I | NL | N/A | N/A | ✓ |
| ChemAgent | I | SL | N/A | N/A | ✓ |
| ChemRAG | E | NL | × | × | × |
| **ChemisTRAG** | **I+E** | **SL** | **SL** | ✓ | ✓ |

**Knowledge Sources and Integration.** Methods for equipping LLMs with chemical knowledge can be divided into those using internal and external knowledge. Internal knowledge methods equip LLMs with chemical expertise through training on domain-specific data (Zhang et al., 2024; Yu et al., 2024; Fang et al., 2024). However, updating knowledge in these approaches requires costly model retraining. External knowledge methods incorporate outside sources through tool augmentation (Bran et al., 2023; M. Bran et al., 2024) or retrieval augmentation (Zhong et al., 2025).

**Retrieval Methods.** Focusing on retrieval augmentation, RAG systems like ChemRAG (Zhong et al., 2025) offer a promising solution for integrating up-to-date external knowledge. However, they struggle with chemical terminology due to their reliance on natural language retrieval, which is sensitive to synonym variations and requires domain-specific knowledge to effectively link equivalent chemical expressions. Current retrieval methods for chemistry tasks are primarily text-based.

**Reasoning Optimization.** Additionally, reasoning-optimized frameworks like StructChem (Ouyang et al., 2024) and ChemAgent (Tang et al., 2025) are designed to fully leverage LLMs' internal knowledge by structuring the reasoning steps or dynamically building knowledge bases. While they improve reasoning, they do not integrate external knowledge augmentation.

ChemisTRAG integrates innovations across all three aspects: (1) a tabular knowledge base leveraging external knowledge; (2) a table-based retriever transforming queries into structured tuples for precise retrieval, supporting diverse chemical names; and (3) an adaptive reasoner grounding generation on evidence while utilizing LLMs' internal knowledge for step-by-step inference.

## 3 DATA CONSTRUCTION

Figure 2 illustrates the data construction workflow, with the left side depicting KB development and the right side showing benchmark data preparation.

**Structural Knowledge Base.** We construct a table KB as the foundation for our RAG system. Following the paradigm of established chemical datasets (Lowe, 2017), we begin by collecting chemical patents from the official website of the USPTO. To ensure data freshness and mitigate leakage risks, we use patents granted between 2020 and 2025. This process yields 85,650 patents, which are further divided into 431,634 text snippets based on line breaks.

To balance efficiency with accuracy, we employ a two-stage LLM pipeline for automatic extraction and verification of reaction information. In the first stage, we use Qwen-3-8B to extract key reaction information (e.g., reactants, products, solvents, catalysts, conditions) from patent snippets. These attributes are organized into a tabular format, with each row representing a single reaction and each column an attribute, yielding an initial set of 205,773 reactions. Subsequently, we employ a more powerful LLM, GPT-OSS-20B, to validate the chemical feasibility of each extracted reaction. This verification step maintains the tabular structure while ensuring data reliability, reducing the dataset to 80,663 validated reactions.

Finally, we query three public compound databases (PubChem, ChEBI, and OPSIN) to enrich each compound with detailed metadata, including IUPAC names, SMILES strings, descriptions, and molecular weights. These metadata are stored in a complementary compound table, where each row represents a unique compound with its attributes. Reactions containing compounds not found in these databases are flagged as invalid and filtered out. Ultimately, this pipeline results in a struc-

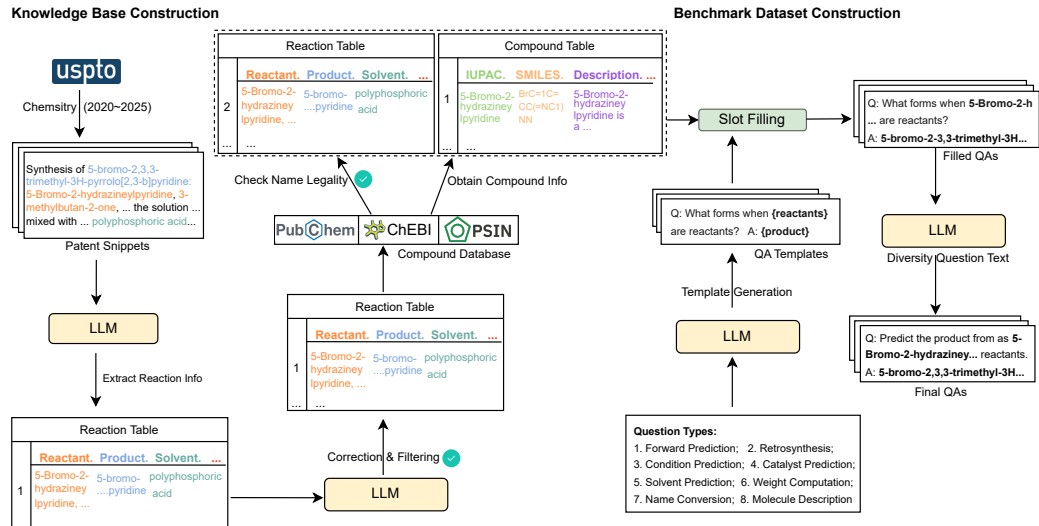

Figure 2: The construction of knowledge base and evaluation data.

tured KB containing 23,105 valid reactions and 38,495 unique compounds. Appendix A shows the statistics of this data.

**Evaluation Data.** Existing reaction evaluation datasets (Guo et al., 2023; Zhang et al., 2024; Yu et al., 2024) are often constructed based on chemical patent data (USPTO) prior to 2016 (Schneider et al., 2016; Lowe, 2017; Jin et al., 2017; Liu et al., 2017). Such data is not timely and risks data leakage, compromising the fairness and reliability of LLM evaluation. To address this, we build our benchmark data using the aforementioned KB. This approach leverages the KB's up-to-date data and enables explicit mapping between benchmark samples and KB entries, supporting quantitative analysis of how retrieval enhances LLM performance on chemical tasks.

We first define eight task types for chemistry question-answering, with references to task designs from existing evaluation datasets (Guo et al., 2023; Zhang et al., 2024; Fang et al., 2024) to ensure comprehensive assessment of LLM capabilities. Among these, five are reaction-centric tasks: product prediction, reactant prediction, solvent prediction, catalyst prediction, and reaction condition prediction. These tasks collectively cover key aspects of reaction analysis. The remaining three are compound-centric tasks: molecular weight calculation, name conversion, and compound description. Appendix A presents the justification of these task choices.

Given the widespread adoption and strong performance of LLMs in question generation (Guo et al., 2024), we use an LLM to generate questions. We use GPT-OSS-20B for template creation instead of direct question generation to ensure accuracy and minimize hallucinations. Specifically, we input each task type to GPT-OSS-20B and prompt it to generate 20 natural language question templates per task. Each template includes slots for task-specific information (e.g., "What is the catalyst required for the reaction involving [reactant 1] and [reactant 2]?").

For question instantiation, we sample 600 entries from the reaction table to populate templates for reaction-centric tasks, and 600 entries from the compound table for compound-centric tasks. This process generates 4,800 initial QA pairs. To enhance linguistic diversity and realism, we use GPT-OSS-20B to paraphrase these questions under strict constraints to preserve their original meaning and factual content. Finally, we standardize answers into concise formats such as short names, lists, or numbers, which are directly extracted from the KB tables. This standardization facilitates quantitative evaluation. This process yields 4,800 high-quality, diverse, multi-task QA pairs with traceable origins, as each pair can be linked to a specific entry in the KB.

We conduct a human evaluation with two chemistry PhD students on 240 samples. Each annotator independently judge whether (1) Naturalness: the question is natural and aligned with typical human user query style, and (2) Correctness: the answer correctly and completely addresses the question. An item is accepted only if both criteria are satisfied. The sampled data shows a pass rate of 95.8%

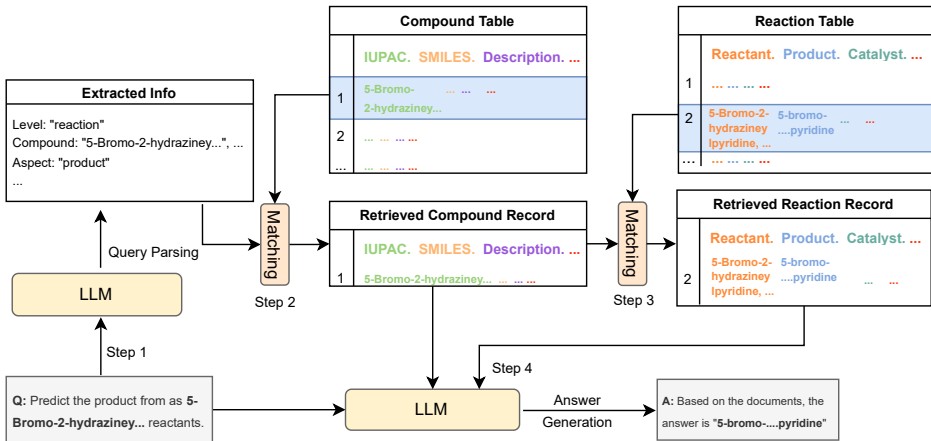

Figure 3: The pipeline of query parsing and knowledge retrieval.

with a high agreement (97.5% of cases receiving matching judgments) between the two experts, demonstrating a high reliability of our constructed evaluation data.

## 4 METHODOLOGY

Building on our structured KB, we now present ChemisTRAG's retrieval and reasoning parts, designed to overcome the dual challenges of semantic matching in chemical queries and the accuracy-generalization trade-off in RAG. The framework integrates a table-based retriever for precise information access and an adaptive reasoner for robust inference.

### 4.1 TABLE-BASED RETRIEVER: STRUCTURED PROJECTION PARADIGM

The unstructured nature of natural language queries, often containing redundant or varied terminology, poses a direct challenge for precise alignment with our tabular KB. To bridge this semantic gap, we introduce a *structured projection* paradigm that transforms queries into schema-aligned representations, as illustrated in Figure 3.

We first map natural language queries to a structured tuple representation:

$$T_q = f_{\text{parse}}(q) = \langle e_{\text{type}}, e_{\text{target}}, a_{\text{query}} \rangle \tag{1}$$

Here, $T_q$ denotes the structured query tuple, and $f_{\text{parse}}$ is an LLM instructed by a parsing prompt $P_{\text{parse}}$ that extracts the tuple components from query $q$. The tuple consists of: $e_{\text{type}} \in \{\text{compound}, \text{reaction}\}$ indicating the entity type; $e_{\text{target}}$ representing the target entity (e.g., SMILES or IUPAC string); and $a_{\text{query}}$ denoting the query intent (e.g., reaction product inquiry).

Based on $e_{\text{type}}$, we perform schema-aligned retrieval against our KB $\mathcal{K} = (C, R)$, where $C = \{c_i\}_{i=1}^m$ is the compound table and $R = \{r_j\}_{j=1}^n$ is the reaction table.

For compound-focused queries, we define a matching function $\text{match}_{\text{comp}} : e_{\text{target}} \times C \rightarrow \mathbb{R}$ that computes relevance scores using ROUGE-L similarity for string overlap measurement. The retrieval selects the top-$k$ most relevant entries:

$$S_C = \underset{S \subseteq C, |S|=k}{\arg\max} \sum_{c_i \in S} \text{match}_{\text{comp}}(e_{\text{target}}, c_i) \tag{2}$$

Notably, the tabular structure of our KB ensures that different representations of the same compound (e.g., common name, IUPAC name, SMILES) are consolidated within a single entry. The matching function compares the query's $e_{\text{target}}$ against the appropriate structured fields in the table, mitigating the synonym problem at the retrieval stage.

For reaction-focused queries, retrieval proceeds through relational mapping by first identifying the top-$k$ most relevant compounds $C_{\text{rel}}$ as above. Let $R_{\text{linked}} = \{r_j \in R \mid \exists c_i \in C_{\text{rel}} : \text{link}(c_i, r_j)\}$ be the set of reactions associated with these compounds, where link represents the compound-reaction

relationship in our KB. We then select the top-$k$ reactions from $R_{\text{linked}}$ based on the maximum matching score of their constituent compounds:

$$C_{\text{rel}} = \underset{S \subseteq C, |S|=k}{\arg\max} \sum_{c_i \in S} \text{match}_{\text{comp}}(e_{\text{target}}, c_i) \tag{3}$$

$$S_R = \text{top-k}_{r_j \in R_{\text{linked}}} \left( \max_{c_i \in C_{\text{rel}} \text{ s.t. link}(c_i, r_j)} \text{match}_{\text{comp}}(e_{\text{target}}, c_i) \right) \tag{4}$$

The retrieved set $S$ (either $S_C$ or $S_R$) serves as the foundational context for subsequent reasoning, converted to structured string format for compatibility.

## 4.2 Adaptive Reasoner: Decoupled Inference Framework

A real-world challenge for RAG systems is balancing faithful extraction from retrieved evidence with the LLM's inherent reasoning capability (Dai et al., 2024; Yan et al., 2024). An over-reliance on retrieval can lead to errors when evidence is incomplete, noisy, or irrelevant, whereas excessive dependence on intrinsic reasoning may ignore relevant context (Chen et al., 2024; Wang et al., 2024a). Our method addresses this through a *decoupled inference framework* (Figure 4) that separates logical planning from factual grounding. This maintains coherent reasoning by integrating retrieved entries with the LLM's capabilities.

We formalize the reasoning process as a composition of three specialized functions: **planning** (constructing the reasoning structure), **grounding** (integrating retrieved evidence), and **execution** (synthesizing the final answer).

**Reasoning Prototype Generation (Plan).** The planning function $f_{\text{plan}}$ constructs a reasoning prototype that captures the problem-solving logic independent of specific evidence:

$$R_{\text{prototype}} = f_{\text{plan}}(q) \tag{5}$$

This function is implemented as an LLM guided by a planning prompt $P_{\text{plan}}$ that generates candidate reasoning chains structured as sequential steps, forming a logical skeleton for subsequent grounding.

**Evidence Grounding.** The grounding function $f_{\text{ground}}$ integrates retrieved entries into the reasoning prototype:

$$R_{\text{grounded}} = f_{\text{ground}}(R_{\text{prototype}}, S) \tag{6}$$

Analogously, this function employs an LLM with a grounding prompt $P_{\text{ground}}$ to adapt the generic reasoning plan to the specific evidence contained in retrieval set $S$ to ensure factual accuracy.

**Answer Synthesis (Execute).** Finally, the execution function $f_{\text{execute}}$ synthesizes the grounded reasoning with query intent to produce the final answer:

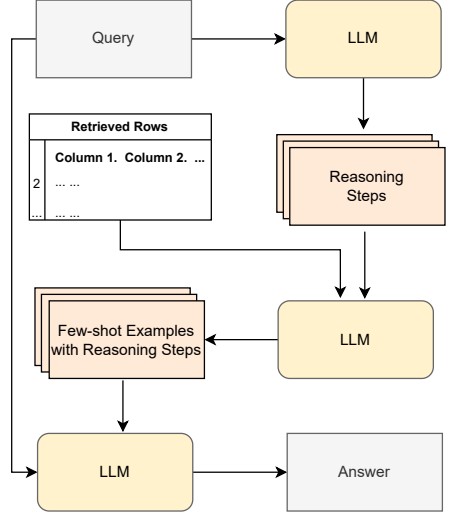

Figure 4: Our adaptive reasoning method.

$$A = f_{\text{execute}}(q, R_{\text{grounded}}) \tag{7}$$

Similarly, $f_{\text{execute}}$ leverages an execution prompt $P_{\text{execute}}$ to generate the final output based on the grounded reasoning chain.

## 5 Experiments

**Evaluation Setup.** We evaluate model performance on eight chemical tasks: Product Prediction (PP), Reactant Prediction (RP), Condition Prediction (CoP), Solvent Prediction (SP), Catalyst Prediction (CaP), Weight Computation (WC), Name Conversion (NC), and Molecule Description (MD). We use the most common chemical naming systems, IUPAC and SMILES, as inputs. The same set of questions is retained across tests, with only the compound naming format varied. For string

Table 2: Performance comparison of RAG methods across different tasks and input types.

| | Reaction | | | | | Compound | | | Overall | Impr. |
|---|---|---|---|---|---|---|---|---|---|---|
| | PP | RP | CoP | SP | CaP | WC | NC | MD | | |
| Overall | | | | | | | | | | |
| w/o RAG | 21.04 | 34.25 | 27.52 | 18.39 | 17.52 | 28.50 | 24.46 | 28.57 | 25.02 | - |
| w/ TextRAG | 28.86 | 36.27 | 44.42 | 43.15 | 29.51 | 51.17 | 42.46 | 45.24 | 40.14 | 15.11 |
| w/ ChemisTRAG | **50.54** | **52.55** | **57.98** | **66.48** | **57.80** | **91.17** | **82.11** | **78.89** | **67.19** | **42.16** |
| IUPAC | | | | | | | | | | |
| w/o RAG | 34.03 | 40.64 | 28.86 | 26.83 | 24.20 | 34.67 | 32.31 | 33.50 | 31.88 | - |
| w/ TextRAG | 39.83 | 41.84 | 50.93 | 57.62 | 35.39 | 78.67 | 63.77 | 67.10 | 54.39 | 22.51 |
| w/ ChemisTRAG | **55.19** | **56.53** | **62.64** | **74.07** | **64.37** | **94.01** | **78.34** | **87.82** | **71.62** | **39.74** |
| SMILES | | | | | | | | | | |
| w/o RAG | 8.05 | 27.86 | 26.17 | 9.95 | 10.84 | 22.33 | 16.61 | 23.63 | 18.18 | - |
| w/ TextRAG | 17.89 | 30.70 | 37.91 | 28.68 | 23.63 | 23.67 | 21.15 | 23.37 | 25.89 | 7.71 |
| w/ ChemisTRAG | **45.89** | **48.57** | **53.32** | **58.89** | **51.23** | **88.33** | **85.88** | **69.96** | **62.76** | **44.58** |

outputs (e.g., molecule prediction, description, NC to IUPAC), we use ROUGE-L to compare with ground truth. For NC to SMILES, we calculate molecular similarity via RDKit. For numerical outputs (e.g., CoP, WC), we compare extracted values with ground truth (allowing decimal tolerance) and count correctness. Final results are presented as averages.

**Implementation Details.** Appendix D presents the implementation details of our experiments.

### 5.1 COMPARISON TO TEXT-BASED RAG

We compare ChemisTRAG against text-based RAG (as in ChemRAG (Zhong et al., 2025)) for chemistry QA, using Qwen-3-8B as the base LLM. TextRAG uses source text paragraphs of our tabular KB as the knowledge source with Qwen-3-8B-Embedding retriever. We evaluate at the reaction-level with five tasks and at the molecule-level with three tasks. We split experiments by two input formats for chemical entities: IUPAC names and SMILES strings. This split helps assess how each RAG method adapts to different chemical naming and representation conventions.

Table 2 presents the comparison results. Direct inference performs poorly across all eight tasks with an overall score of 25.02. This shows LLMs lack sufficient inherent knowledge for complex chemistry reasoning. RAG consistently boosts LLM performance, demonstrating the value of external knowledge integration. TextRAG reaches an overall score of 40.14, bringing an average improvement of 15.11 over direct inference. Our ChemisTRAG outperforms TextRAG significantly, achieving the highest overall score of 67.19 with an average improvement of 42.16 over direct inference.

Performance shows notable differences across IUPAC and SMILES inputs, revealing how input representation affects LLM and RAG effectiveness. Across all methods, models score higher with IUPAC than SMILES.For Direct reasoning, IUPAC achieves an overall score of 31.88, much higher than SMILES at 18.18. This difference likely comes from IUPAC names being more similar to natural language and easier for LLMs to understand. TextRAG widens the performance gap between IUPAC and SMILES. With TextRAG, IUPAC's overall score rises to 54.39 while SMILES only reaches 25.89. This suggests current text-based retrievers lack proficiency in processing chemistry-specific representations. Their failure to align SMILES queries hurts retrieval accuracy and thus limits performance gains.

In contrast, ChemisTRAG narrows the gap between the two input formats. For IUPAC, it achieves an overall score of 71.62 while maintaining its lead over TextRAG. For SMILES, it reaches 62.76, a score that is far closer to its IUPAC performance than TextRAG's corresponding gap. This advantage stems from ChemisTRAG's table-based design. By storing chemical information in structured tables, it enables the retriever to align queries with knowledge across different naming conventions. Grounded in this more precise retrieved knowledge, the LLM's performance gap across the two input representations is effectively narrowed.

### 5.2 RETRIEVAL RECALL ANALYSIS

We evaluate knowledge retrieval effectiveness using Recall@5 (denoted as R) and its correlation with model performance scores (denoted as S). The analysis covers two task categories Reaction

and Compound along with two input formats IUPAC and SMILES. Results are presented in Table 3. Recall@5 and performance scores show a strong positive correlation with a Pearson coefficient of 0.96. This demonstrates that retrieval quality directly impacts reasoning correctness.

Compound tasks often achieve higher recall than reaction tasks for both methods. For example, TextRAG achieves 41.22 for Compound tasks versus 23.20 for Reaction tasks. A possible reason is that reaction tasks involve multiple entities and conditions, making relevant knowledge harder to capture. In contrast, compound tasks focus on single molecular properties, so their information is more easily retrievable.

TextRAG shows uniformly low recall values that rarely exceed 42. It also has a large gap in recall score between IUPAC and SMILES inputs. For Reaction tasks TextRAG's Recall@5 for IUPAC is 33.67 nearly three times the 12.73 for SMILES. For Compound tasks the gap remains substantial with 68.67 for IUPAC versus 13.78 for SMILES. Text-based retrievers rely on natural language so they handle IUPAC well due to its language-like structure but struggle with SMILES which uses chemistry-specific symbols.

Table 3: Comparison of Recall@5 and performance on two tasks.

|  | Reaction | | Compound | |
| --- | --- | --- | --- | --- |
|  | R | S | R | S |
| Overall | | | | |
| TextRAG | 23.20 | 36.44 | 41.22 | 46.28 |
| ChemisTRAG | **68.20** | **57.07** | **88.72** | **84.06** |
| IUPAC | | | | |
| TextRAG | 33.67 | 45.13 | 68.67 | 69.85 |
| ChemisTRAG | **76.80** | **62.56** | **89.89** | **86.72** |
| SMILES | | | | |
| TextRAG | 12.73 | 27.76 | 13.78 | 22.72 |
| ChemisTRAG | **59.60** | **51.58** | **87.56** | **81.39** |

ChemisTRAG achieves far higher Recall@5 and narrows the Recall@5 gap between IUPAC and SMILES. For Reaction tasks the gap between IUPAC (76.80) and SMILES (59.60) is under 20. For Compound tasks the gap is minimal with 89.89 for IUPAC and 87.56 for SMILES. ChemisTRAG's table-based design enables retrieval through explicit entity matches. This allows effective mapping of input formats without requiring the retriever to have knowledge of chemical symbols. Its superior Recall@5 explains the higher performance. More relevant retrieved knowledge supports accurate reasoning by the LLM. This lets ChemisTRAG outperform TextRAG across all subcategories by addressing the core retrieval bottleneck of text-based RAG systems.

## 5.3 ABLATION STUDY AND OUT-OF-CORPUS RESULTS

We conduct an ablation study with five method variants to test key components of our approach. We also design two settings to simulate real-world RAG scenarios. The five variants include full ChemisTRAG and four ablated versions. "w/o Prototype" removes the initial reasoning prototype generation. "w/o Adaptation" cuts the step that adapts reasoning to retrieved results. "CoT" (Chain of Thought, Wei et al. (2022)) drops the first two steps and lets LLMs reason step-by-step on retrieved results. "w/o All" uses retrieved results for direct reasoning with no extra steps. The two evaluation settings are In-Corpus (IC) and Out-of-Corpus (OOC). IC tests questions with exact answers in the knowledge base. It checks if LLMs can extract information accurately when relevant data exists. OOC targets questions without exact answers. It tests LLMs' ability to reason with imperfect information. We implement OOC by removing knowledge base entries matching benchmark data. Results are in Figure 5.

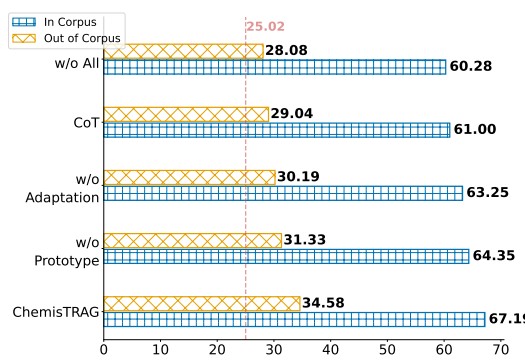

Figure 5: Performance comparison of reasoning variants across IC and OOC RAG scenarios.

All ablated variants perform worse than full ChemisTRAG, showing the necessity of each component of the reasoning framework. "w/o Adaptation" (removing reasoning adaptation) drops performance more than "w/o Prototype" (removing reasoning prototype generation). Probably because it removes the injection of retrieved info into reasoning paths, the model easily produces biased outputs without using retrieval context. CoT outperforms "w/o All" slightly, which indicates that step-by-step reasoning without pre-generated reasoning context gives small benefits in RAG.

Table 4: Performance of various LLMs across direct answering and retrieval methods.

|  | Qwen3-8B | Llama-3.1-8B | ChemLLM-7B | GPT-4o | Qwen3-8B-Think | Deepseek-R1 |
|---|---|---|---|---|---|---|
| w/o RAG | 25.02 | 20.59 | 15.92 | 36.71 | 26.43 | 35.90 |
| w/ BM25 | 57.22 | 48.34 | 24.57 | 58.64 | 57.83 | 55.29 |
| w/ Qwen3 Emb | 40.47 | 35.61 | 27.43 | 41.48 | 40.35 | 40.39 |
| w/ T-Retriever | **67.19** | **59.05** | **30.17** | **72.06** | **69.53** | **73.43** |

All variants perform much worse in OOC than IC. This shows accurate knowledge and precise retrieval are key for good reasoning. Even so, ChemisTRAG still boosts LLM performance in OOC. Its OOC score of 34.58 is higher than the 25.02 baseline. This improvement proves our method lets models learn from context. They can do analogical reasoning when exact retrieval fails. It shows the method's adaptability. Ablation trends in OOC match those in IC. This consistency confirms each component works well whether exact knowledge exists or not.

## 5.4 RETRIEVAL STRATEGY AND MODEL ANALYSIS

We assess how different retrieval paradigms affect performance across diverse LLMs. Using our tabular KB, we test six models: general open-source (Qwen3-8B, Llama-3.1-8B (Grattafiori et al., 2024)), commercial (GPT-4o (Achiam et al., 2023)), chemistry-specialized (ChemLLM-7B (Zhang et al., 2024)), and reasoning (Qwen3-8B-Think Mode, Deepseek-R1 (Guo et al., 2025)). We compare three retrieval approaches: statistics-based (BM25 (Robertson et al., 2009)), vector-based (Qwen-3-embedding (Zhang et al., 2025)), and our table-based retriever (T-Retriever).

**Influence of Retrieval Strategy.** Table 4 shows how knowledge retrieval enhances performance across diverse LLMs, compared to direct reasoning without retrieval. All retrieval approaches outperform direct reasoning for every LLM, proving RAG consistently boosts chemistry problem-solving. Notably, the vector similarity method using Qwen3 Embedding, a state-of-the-art encoder, performs worse than the statistics-based BM25 across most models. This gap reveals current embedding models lack proficiency in chemistry-specific knowledge retrieval. Our table-based retrieval achieves the highest scores across all LLMs.

**Performance on Different Models.** Retrieval effectiveness on performance aligns with the LLM's inherent capabilities. Commercial models like GPT-4o and reasoning-focused models such as Deepseek-R1 deliver the best overall performance. They leverage retrieved knowledge well with their strong base reasoning. General open-source models show moderate gains. The chemistry-specialized ChemLLM-7B lags however. Its specialized fine-tuning likely reduces in-context learning flexibility, limiting retrieval benefits. Our method delivers the most significant gains across all model types, showing its universal effectiveness.

**Comparison with Domain-Specific Encoders.** We further examine whether domain-specific dense retrievers offer advantages over lexical matching for entity alignment in our table-based retrieval. We compare our default ROUGE-L with ChemBERTa (Chithrananda et al., 2020), a specialized chemistry encoder. Table 5 reveals an interesting trade-off: ROUGE-L significantly outperforms ChemBERTa in IC settings, confirming that precise lexical linking is superior when

Table 5: Comparison of lexical and domain-specific encoders within T-Retriever.

| Setting | Encoder | Reac. | Comp. | Overall |
|---|---|---|---|---|
| IC | ChemBERTa | 48.38 | 74.74 | 58.27 |
| | ROUGE-L | **57.07** | **84.06** | **67.19** |
| OOC | ChemBERTa | **39.72** | 33.20 | **37.27** |
| | ROUGE-L | 30.22 | **41.85** | 34.58 |

exact knowledge exists. However, in OOC settings, ChemBERTa performs better on Reaction tasks. This suggests that dense embeddings capture semantic relevance (e.g., similar reaction types), which aids analogical reasoning when exact matches are missing.

## 5.5 GENERALIZABILITY AND ROBUSTNESS

To verify the generalization of our method beyond the constructed dataset, we evaluate Chemis-TRAG on two external public benchmarks: ChemBench (Zhang et al., 2024) and SciBench (Wang et al., 2024b). These datasets cover diverse tasks including yield prediction and college-level chemistry problems. As shown in Table 6 (Left), ChemisTRAG consistently outperforms both the direct

Table 6: Performance on public benchmarks (Left) and robustness against input perturbation (Right).

| | External Benchmarks | | Perturbed Subset | | |
| | ChemBench | SciBench | Reaction | Compound | Overall |
|---|---|---|---|---|---|
| w/o RAG | 55.38 | 30.57 | 21.11 | 24.27 | 22.30 |
| w/ TextRAG | 57.63 | 32.75 | 34.21 | 44.16 | 37.94 |
| w/ ChemisTRAG | **60.25** | **34.93** | **53.33** | **79.95** | **63.31** |

inference baseline and the text-based ChemRAG. This demonstrates that our structured retrieval paradigm generalizes effectively to external datasets and broader chemical tasks.

We further assess the robustness of our system against input noise, a common issue in real-world applications. We introduce random perturbations (e.g., typos, name variations) to the entity names within a 15% random sample of our evaluation dataset. Then, we evaluate performance specifically on this perturbed subset. As shown in Table 6 (Right), while performance naturally drops for all methods compared to clean data, ChemisTRAG maintains a significant performance margin. It achieves an overall score of 63.31 on this noisy subset, surpassing Text-based RAG (37.94) and direct inference (22.30), demonstrating strong resilience to entity perturbations.

## 5.6 Impact of the Number of Retrieved Entries

To explore how the quantity of retrieved entries ($K = 1, 3, 5, 10$) affects the overall performance of the RAG system, we investigate two key metrics across different $K$ values: Recall@$K$ and the corresponding performance.

Figure 6 shows a consistent upward trend in both Recall@$K$ and performance scores as $K$ increases. A larger $K$ expands the retrieval scope, enhancing the probability of capturing critical relevant knowledge that supports accurate reasoning. However, this expansion may also introduce redundant or irrelevant information, which may interfere with the LLM's ability to focus on core task-related content. This aligns with a common tradeoff in RAG systems.

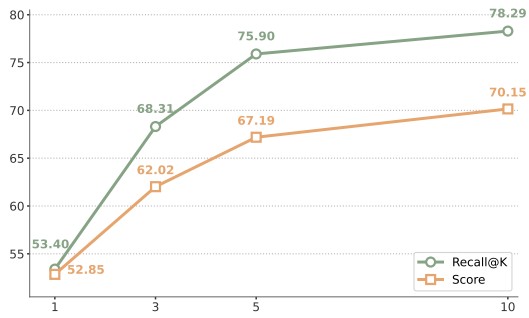

Figure 6: Recall and performance scores across different retrieved numbers for ChemisTRAG.

Notably, the growth rate of both metrics varies across $K$ ranges. The improvement is most pronounced when $K$ increases from 1 to 5. In contrast, when $K$ further increases from 5 to 10, the growth of both Recall@$K$ and performance slows. Considering both performance gains and system efficiency, $K = 5$ should be a good choice.

## 6 Conclusion and Future Work

We proposed ChemisTRAG to enhance LLMs in chemistry problems, which encompasses the full pipeline of RAG driven by tables, namely the table-based KB, structured retriever, and adaptive reasoning method. We also built a multi-task benchmark dataset for a fair and comprehensive evaluation. Experimental results showed the effectiveness of ChemisTRAG. We discuss the limitations of our work and potential future improvement. **1)LLM Hallucination.** While RAG grounds generation to mitigate hallucinations, intrinsic LLM flaws persist. Future work may implement hallucination detection or multi-model checks to further reduce this risk. **2) Multi-step Reactions and Reasoning.** Our current work focuses on single-step QA. Addressing multi-step reactions and mechanistic reasoning is a crucial future challenge. This may require collecting expert-annotated CoT data and exploring methods like Reinforcement Learning to enhance long reasoning.

## ETHICS STATEMENT

All raw corpora used in this study are sourced exclusively from publicly available datasets. During data collection, we strictly adhered to the data access policies and web crawling protocols specified by the original data providers. The processed data and associated code generated in this research are intended solely for academic and research purposes. Although RAG significantly enhances the performance of LLMs, LLMs still have inherent randomness and hallucination issues. Thus, users need to carefully verify the system's outputs, especially when these outputs are intended for chemical synthesis and research.

## REPRODUCIBILITY STATEMENT

We are committed to ensuring the reproducibility of our research. The data and code used in this study will be made publicly available. All experiments were conducted three times, and the average values were reported. All key parameters and settings of experiments are disclosed in this paper.

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

## A  DATASET

**Task Definition and Justification** Our evaluation benchmark is designed around two fundamental units in chemistry: reactions and compounds, which represent the most common entities in chemical research and applications (Han et al., 2025). For reaction-centric tasks, we follow established literature in defining five key tasks that comprehensively cover the reaction process (Guo et al., 2023; Fang et al., 2024; Zhang et al., 2024; Zhong et al., 2025): predicting products from reactants, predicting reactants from products, and predicting key reaction conditions including solvents, catalysts, and specific reaction parameters like temperature and pH value. These tasks encompass the complete reaction workflow from input to output and conditions. We intentionally exclude yield prediction due to inconsistent reporting in patent texts, where yield descriptions often require complex calculations from mass values or refer to intermediate steps, making reliable ground-truth establishment challenging.

For compound-centric tasks, we focus on three fundamental capabilities: name conversion between different chemical representations, basic chemical calculations such as molecular weight, and molecular description generation. These tasks assess LLMs' understanding of chemical entities. A key advantage of our benchmark design is that each question-answer pair is explicitly traceable to specific entries in our knowledge base. This traceability enables precise analysis of how retrieval quality impacts LLM performance, particularly in distinguishing cases where exact matches are available versus those requiring reasoning with similar information. The dataset construction process ensures both coverage of essential chemistry tasks and reliable evaluation of retrieval-augmented generation methods.

**Statistics.** Our knowledge base comprises 38,495 unique compounds and 23,105 chemical reactions extracted from USPTO (United States Patent and Trademark Office) patents granted between 2020 and 2025. The compound table includes detailed metadata, with compounds having an average molecular weight of 223.41 and each compound participating in approximately 1.87 reactions on average. Only 12.14% of compounds contain textual descriptions, with an average length of 34.91 words per description. The chemical representations show distinct characteristics: IUPAC names average 53.08 characters in length, while SMILES strings are more concise at 35.21 characters.

The reaction table demonstrates high coverage for core reaction components. Reactants and products are present in 100% of reactions, while reaction conditions are specified in 97.13% of cases. Solvents

are documented in 80.63% of reactions, though catalysts are reported less frequently, appearing in only 25.61% of reactions.

For evaluation, we constructed a benchmark of 4,800 question-answer pairs evenly distributed across eight tasks: five reaction-centric and three compound-centric. Each task contains 600 instances, with half using IUPAC names and half using SMILES strings as input to ensure fair comparison across representation formats. Questions average 17.51 words in length, while answers are more concise at 6.89 words, reflecting the focused nature of chemical question answering. This balanced design enables comprehensive assessment of retrieval-augmented generation methods across diverse chemical reasoning scenarios.

**Data Examples.** To intuitively reflect the dataset structure, Tables 7, 8, and 9 show representative examples of compounds, reactions, and QA pairs, respectively. For Tables 7 (compounds) and 8 (reactions), considering their more attributes and the need for conciseness and space saving, attributes are arranged vertically (listed row-wise on the left). This layout avoids overly wide tables and facilitates cross-sample comparison. Table 9 (QA pairs) retains the original column-wise attribute arrangement due to fewer attributes, ensuring readability while aligning with the document format.

Table 7: Compounds Data Examples

| Compound Attribute | Compound 1 (mol_id: 2) | Compound 2 (mol_id: 21853) |
|---|---|---|
| mol_id | 2 | 21853 |
| iupac | triphenylphosphane | diethyl benzene-1,4-dicarboxylate |
| smiles | C1=CC=C(C=C1)P(C2=CC=...) | CCOC(=O)C1=CC=C(C=C1)C(=O)... |
| common_name | triphenylphosphine | diethyl terephthalate |
| molecular_formula | C18H15P | C12H14O4 |
| molecular_weight | 262.3 | 222.24 |
| relevant_rxn | 9661,10584,... | - |
| pubchem_id | 11776 | 12483 |
| description | Triphenylphosphine is a member of the class of tertiary phosphines... | - |

# B  ALGORITHM OF CHEMISTRAG

Algorithm 1 provides the procedural view of ChemisTRAG's answer generation process based on the tabular KB, summarizing the end-to-end workflow from query processing to answer generation. The algorithm integrates the structured projection retrieval and adaptive reasoning components described in Section 4.

The algorithm formalizes the three-stage pipeline of ChemisTRAG. Steps 1-2 handle query parsing into structured tuples. Steps 4-9 implement the table-based retrieval, with compound and reaction queries following different paths. Steps 11-13 encapsulate the adaptive reasoning process, where each function ($f_{\text{plan}}$, $f_{\text{ground}}$, $f_{\text{execute}}$) represents an LLM operation guided by specific prompts as described in Section 4.2.

Table 8: Reactions Data Examples

| Reaction Attribute | Reaction 1 (ID: 1) | Reaction 2 (ID: 2868) |
|---|---|---|
| Reaction ID | 1 | 2868 |
| Patent | US-20240317765-A1 | US-20240018095-A1 |
| Reaction Description | [0762] When a halogenation reaction of a hydroxy group is carried out in each step, examples of the halogenating agent include hydrohalic acids and acid halides of inorganic acids... | EXAMPLES [0042] Synthesis of stearic acid amide (SAA) from SA and EDA: 4.74 g of methyl stearate and 2.86 g of EDA were combined in a sealed 20 mL vial... |
| Reactants | ethanol.triphenylphosphine.carbon tetrachloride | methyl octadecanoate.ethylenediamine |
| Products | 1-chloroethane.triphenylphosphine oxide | octadecanamide |
| Solvents | - | ethyl acetate |
| Catalysts | - | - |
| Operations | reacting | heating.filtration |
| Conditions | {'temperature': 0} | {'temperature': '90', 'time': '67'} |
| Yield | 93 | 90.0 |
| Notes | Appel halogenation of ethanol to 1-chloroethane with triphenylphosphine and carbon tetrachloride, yielding triphenylphosphine oxide as by-product. | The product was obtained as white powder. |

Table 9: QA Pairs Data Examples (4 Samples)

| rxn_id | mol_id | question | answer | qa_type | input_type |
|---|---|---|---|---|---|
| 22683 | - | If the reactants are cumene hydroperoxide and cumyl alcohol (IUPAC), what main product forms? | propylene oxide | product_prediction | iupac |
| 22683 | - | What's the reaction product of [O-]O.C1(=CC=CC=C1)C(C)C and C(C)(C)(C1=CC=CC=C1)O? | propylene oxide | product_prediction | smiles |
| - | 25992 | Calculate the molecular mass for 2-methyl-2-phenylpropanamide. | 163 | mass_prediction | iupac |
| - | 21854 | What's the SMILES for the compound with IUPAC name methyl 4-ethynylbenzoate? | C(#C)C1=CC=C(C(=O)OC)C=C1 | name_conversion | iupac |

---

**Algorithm 1** ChemisTRAG: Table-based RAG for Chemistry QA

---

**Require:** Natural language query $q$, compound table $C$, reaction table $R$
**Ensure:** Final answer $A$

1:
2: **Step 1: Query Parsing**
3: $T_q \leftarrow f_{\text{parse}}(q) = \langle e_{\text{type}}, e_{\text{target}}, a_{\text{query}} \rangle$
4:
5: **Step 2: Schema-aligned Retrieval**
6: **if** $e_{\text{type}} = $ "compound" **then**
7: $\quad S \leftarrow$ top-$k$ $c_i \in C$ by $\text{match}_{\text{comp}}(e_{\text{target}}, c_i)$
8: **else**
9: $\quad C_{\text{rel}} \leftarrow$ top-$k$ compounds matching $e_{\text{target}}$
10: $\quad R_{\text{linked}} \leftarrow \{r_j \in R \mid \exists c_i \in C_{\text{rel}} : \text{link}(c_i, r_j)\}$
11: $\quad S \leftarrow$ top-$k$ $r_j \in R_{\text{linked}}$ by max compound similarity
12: **end if**
13:
14: **Step 3: Adaptive Reasoning**
15: $R_{\text{prototype}} \leftarrow f_{\text{plan}}(q)$ {Generate reasoning prototype}
16: $R_{\text{grounded}} \leftarrow f_{\text{ground}}(R_{\text{prototype}}, S)$ {Ground with retrieved evidence}
17: $A \leftarrow f_{\text{execute}}(q, R_{\text{grounded}})$ {Synthesize final answer}
18:
19:
20: **return** $A$

---

## C  CASE STUDY

**Case Study 1: Exact Match Scenario.**

---

**Answer Generation based on Exact Match with Table-driven Retrieval**

**Query:** *When benzaldehyde and benzohydrazide react, what most possible product is generated?*

**Ground Truth:** *N-Benzylidene benzohydrazide*

**Parsed Tuple:**

```
{
    "level": "reaction",
    "compounds": ["benzaldehyde", "benzohydrazide"],
    "aspect": "product"
}
```

**Retrieved Evidence:**

```
{
    "rxn_id": 4900,
    "reactants": "benzaldehyde and benzohydrazide",
    "products": "N-Benzylidene benzohydrazide",
    "solvents": "ethanol and ethyl acetate",
    "catalysts": "N/A",
    "temperature": 25,
    "match_info": "exact"
}
```

**Reasoning Prototype (Step 1 in the Adaptive Reasoner):**

Step 1: Identify the reactants. Benzaldehyde ($C_6H_5CHO$) is an aromatic aldehyde, and benzohydrazide ($C_6H_5CONHNH_2$) is a derivative of hydrazine with a benzoyl group.
Step 2: Recognize the type of reaction. The reaction between an aldehyde and a hydrazide typically involves nucleophilic attack by the hydrazide on the carbonyl carbon of the aldehyde.
Step 3: Determine the mechanism. The hydrazide acts as a nucleophile, attacking the electrophilic carbonyl carbon of benzaldehyde, leading to the formation of a tetrahedral intermediate.
Step 4: Consider the stability of the intermediate. The intermediate is unstable and undergoes further reaction, typically leading to the elimination of water and the formation of a hydrazone.
Step 5: Identify the product. The most likely product is benzohydrazone, which has the structure $C_6H_5CH=N-NH-C_6H_5$. This compound is formed through the condensation of benzaldehyde with benzohydrazide.

**Evidence-Grounded Reasoning (Step 2 in the Adaptive Reasoner):**

Step 1: Identify the reactants. Benzaldehyde ($C_6H_5CHO$) is an aromatic aldehyde, and benzohydrazide ($C_6H_5CONHNH_2$) is a hydrazine derivative with a benzoyl group.
Step 2: Recognize the type of reaction.
The reaction between this aldehyde and hydrazide typically involves nucleophilic attack by the hydrazide on the aldehyde's carbonyl carbon.
Step 3: Determine the mechanism. The hydrazide acts as a nucleophile, attacking the electrophilic carbonyl carbon of benzaldehyde, leading to a tetrahedral intermediate.
Step 4: Consider the stability of the intermediate. The intermediate is unstable, undergoing water elimination to form a conjugated imine - containing product.
Step 5: Identify the product. The most likely product is N-Benzylidene benzohydrazide , formed by condensing benzaldehyde and benzohydrazide.

**Final Answer:** *N-Benzylidene benzohydrazide*

---

We present a case study to exemplify ChemisTRAG's performance in *exact match scenarios*, where query compounds have direct KB correspondences. For clarity, we present the structured KB entries in JSON format, though they are originally stored in relational tables. Our table-based retriever maps chemical names to correct reaction entries via structured tuple parsing, bypassing the semantic matching challenges inherent in text-based approaches. Meanwhile, the adaptive reasoner generates chemically accurate answers by grounding reasoning in retrieved evidence. This case validates our core design principle: table-based retrieval enables precise matching, ensuring reliable performance when exact knowledge is available in the KB.

---

**Adaptive Reasoning with Absent Exact Matches (OOC Scenario)**

**Query:** *Calculate the molecular weight of the compound 2,5-dichloropyrimidine*

**Ground Truth:** *148.98*

**Parsed Tuple:**

```
{
    "level": "compound",
    "compounds": ["2,5-dichloropyrimidine"],
    "aspect": "weight"
}
```

**One of Retrieved Entries (Not Exactly Matching):**

```
{
    "mol_id": 1813,
    "iupac": "4,5-dichloropyrimidine",
    "smiles": "C1=C(C(=NC=N1)Cl)Cl",
    "common_name": "4,5-dichloropyrimidine",
    "molecular_formula": "C4H2Cl2N2",
    "molecular_weight": 148.98,
    "description": nan,
    "match": "similar"
}
```

**Reasoning Prototype (Step 1 in the Adaptive Reasoner):**

Step 1: Identify the molecular formula of 2,5-dichloropyrimidine. The compound 2,5-dichloropyrimidine has the molecular formula $C_4H_4N_2Cl_2$ .

Step 2: Determine the atomic weights of each element involved. - Carbon (C): 12.01 g/mol - Hydrogen (H): 1.008 g/mol - Nitrogen (N): 14.01 g/mol - Chlorine (Cl): 35.45 g/mol

Step 3: Calculate the contribution of each element to the molecular weight. - Carbon: 4 atoms × 12.01 g/mol = 48.04 g/mol - Hydrogen: 4 atoms × 1.008 g/mol = 4.032 g/mol - Nitrogen: 2 atoms × 14.01 g/mol = 28.02 g/mol - Chlorine: 2 atoms × 35.45 g/mol = 70.90 g/mol

Step 4: Sum the contributions to obtain the molecular weight. Molecular weight = 48.04 g/mol + 4.032 g/mol + 28.02 g/mol + 70.90 g/mol = 150.99 g/mol.

**Evidence-Grounded Reasoning (Step 2 in the Adaptive Reasoner):**

Step 1: Identify the molecular formula of 4,5-dichloropyrimidine. The compound 4,5-dichloropyrimidine has the molecular formula $C_4H_2N_2Cl_2$ .

Step 2: Determine the atomic weights of each element involved. - Carbon (C): 12.01 g/mol - Hydrogen (H): 1.008 g/mol - Nitrogen (N): 14.01 g/mol - Chlorine (Cl): 35.45 g/mol

Step 3: Calculate the contribution of each element to the molecular weight. - Carbon: 4 atoms × 12.01 g/mol = 48.04 g/mol - Hydrogen: 2 atoms × 1.008 g/mol = 2.016 g/mol - Nitrogen: 2 atoms × 14.01 g/mol = 28.02 g/mol - Chlorine: 2 atoms × 35.45 g/mol = 70.90 g/mol

Step 4: Sum the contributions to obtain the molecular weight. Molecular weight = 48.04 g/mol + 2.016 g/mol + 28.02 g/mol + 70.90 g/mol = 148.98 g/mol.

**Final Answer:** *148.98*

---

**Case Study 2: Reasoning with No Exact Matches.** This case demonstrates ChemisTRAG in *out-of-corpus scenarios*, where there is no corresponding entry in the table KB for the given question. The initial reasoning prototype incorrectly calculated the molecular weight using an erroneous formula ($C_4H_4N_2Cl_2$ with a result of 150.99 g/mol). However, the reasoning skeleton is reasonable. The grounding stage adapted the reasoning details by integrating the retrieved similar compound (4,5-dichloropyrimidine, $C_4H_2N_2Cl_2$), whose structural similarity to the query compound enabled accurate weight calculation (148.98 g/mol). The execution stage infers the answer based on this context, resulting in the final answer of 148.98, which matches the ground truth. This exemplifies how our decoupled inference framework maintains reasoning integrity even with imperfect initial plans, leveraging chemical analogies when exact matches are unavailable.

## D IMPLEMENTATION DETAILS

We used the default temperature setting and empirically set top_p to 0.4, ensuring the stability of the LLMs' output while retaining diversity. The number of retrieved entries is set to 5. To achieve rapid inference and memory efficiency, we employed the vLLM library (Kwon et al., 2023) to deploy our open-source LLM calling services. We used the default template of each LLM for LLM prompting. All computational experiments were conducted on a server equipped with $8 \times$ L20 48G GPUs. For the inference of models with 7∼8B size, a single GPU was used; 4 GPUs were used for the inference of GPT-OSS-20B model. For larger-scale and commercial LLMs, we utilized the generative AI services hosted by our institution. Supported by vLLM, it took an average of 3.2 seconds to answer a single query using ChemisTRAG on Qwen-3-8B.

---

**Prompt for Reaction Extraction**

You are a chemistry expert who collects specific chemical reactions from texts, targeted to construct a knowledge base.
Your task is to extract chemical reaction information from chemical patents:
1. Reactants (list): Starting materials that directly participate in the chemical reaction process;
2. Products (list): Substances generated after the chemical reaction reaches completion;
3. Solvents (list): Liquids that dissolve reactants or catalysts without being consumed;
4. Catalysts (list): Substances that accelerate reaction rate without being consumed;
5. Condition (dict): Key environmental parameters that affect reaction progress;
6. Remark (str): Additional relevant details not covered by the above items.
Constraints:
- Reactants, Products, Solvents, and Catalysts should be only specific and legitimate compound names (IUPAC), excluding any references such as 'Compound A', 'Compound I', etc., states of matter such as 'solution', 'solid', and any additional descriptions such as 'complex', 'composite';
- Conditions should be a dict with JSON format, where the keys are "temperature", "pH value", and the values should be in the simplest form, containing only numerical values.
- If encounter any parts that cannot be revised to meet the requirements, directly return an empty dict or list.
- Output the corrected dict in JSON format.

---

Figure 7: Prompt for extracting reaction information for chemical patents.

---

**Prompt for Reaction Check**

You are a chemistry expert who collects specific chemical reactions from texts, targeted to construct a knowledge base.
Your task is to check and ensure that each field meets the requirements:
1. Reactants, Products, Solvents, and Catalysts should be only specific and legitimate compound names (IUPAC), excluding any references such as 'Compound A', 'Compound I', etc., states of matter such as 'solution', 'solid', and any additional descriptions such as 'complex', 'composite';
2. Conditions should be in the simplest form, containing only numerical values.
3. The reaction should be reasonable. Use your professional knowledge to revise the incorrect parts, making the reaction formula conform to chemical principles.
Constraints:
- If encounter any parts that cannot be revised to meet the requirements, directly return an empty dict.
- Output the corrected dict in JSON format, consistent with the original reaction info dict.

---

Figure 8: Prompt for checking reaction information.

## E STATEMENT OF GENERATIVE AI USE

We use GPT-4o and DeepSeek for the purpose of correcting grammar, enhancing expressions, and assisting programming. In our research, we employed generative models for the data construction of our knowledge base and evaluation data, which constitute one of our central focuses aiming to enhance model capabilities through the knowledge base and assess such improvement. However, it is crucial to clarify that: (1) The method and experiment are designed by us independently. (2) All experimental datasets were derived from empirical results.

**Prompt for Generation of Question Templates**

You are a chemistry expert. Given a question type, generate 20 chemical questions with different linguistic styles or syntactic structures. Below are the question types along with corresponding inputs and expected outputs:
- Product Prediction: Given reactants, predict the products.
- Reactant Prediction: Given products, predict the reactants.
- Condition Prediction: Given a chemical equation including reactants and products, predict the reaction conditions (e.g., temperature or pH value; specify one).
- Solvent Prediction: As above, predict the solvents.
- Catalyst Prediction: As above, predict the catalyst.
- Weight Computation: Given a compound, calculate its mass.
- Name Conversion: Given a compound name in one format (e.g., IUPAC or SMILES), convert it to another format.
- Molecule Description: Given a compound, provide the corresponding description.
Requirements:
- Simulate real users' queries as much as possible.
- Reserve one placeholder in each question, enclosed in "{}", e.g., "{products}".

Figure 9: Prompt for question template generation.

**Prompt for Question Diversification**

You are a chemistry expert. Given a question generated by an LLM, please rewrite it into a query that is closer to the tone of real users.
Requirements:
- Make as many variations as possible, but only adjust the linguistic style while strictly preserving the core meaning of the query. - Output the rewritten sentence directly.

Figure 10: Prompt for diversifying questions.

**Prompt for Direct Inference**

You are an intelligent assistant. Answer query based on the given few-shot examples.
# Constraints
- If you don't know the answer, make the best guess based on your knowledge.
- Output must be JSON with 'thinking' and 'answer', where 'thinking' is your thinking process, and 'answer' should directly answer the given query in one or a few words.
- 'answer' should specify the compound name or form numeric answer. For question requires you to transform something into SMILES, the 'answer' should output SMILES format.

Figure 11: Prompt for direct inference.

**Prompt for Query Parsing**

You are a chemistry expert. Given a chemical query, extract information into a JSON string with keys:
- 'level': the level involved in the query ('compound' or 'reaction')
- 'compounds': list of compound names in the query
- 'format': input format of compounds in the query ('smiles' or 'iupac'), NOT the output format.
- 'aspect': query focus (e.g., weight, product, reactants, condition, name conversion, etc.) Output only the JSON string.

Figure 12: Prompt for parsing queries to structural information.

**Prompt for Reasoning Prototype Generation**

You are a chemistry expert. Given a chemical query, generate a step-by-step reasoning path to solve it.
Requirements:
1. Each reasoning path must have clear step numbering (e.g., Step 1...;Step 2...;Step 3...).
2. Highlight specific chemical names and numerical values.

Figure 13: Prompt for generating initial reasoning paths.

**Prompt for Reasoning Path Adaptation**

You're a chemistry expert. Adapt the given reasoning path to strictly match the chemical record.
Steps:
1. Start by providing a json string with keys "answer" pertaining to the aspect, copied directly from the corresponding key of the record.
2. Then, replace the compound and reaction information in the reasoning path with data from the record.
Requirements:
1. Strictly extract the name and info from the record without modifications.

Figure 14: Prompt for adapt reasoning paths to retrieval entries.

**Prompt for Final Answer Generation**

You're a chemistry expert. Infer the answer to the query based on the given context.
Requirements:
1. If there is context highly matching the query, you should directly use the answer. Otherwise, take the reasoning paths as few-shot examples and try to find something in common between compounds in the reasoning paths and given query, then infer the answer with step-by-step thinking.
2. If there is no valid context, infer the answer using your knowledge step-by-step.
3. Conclude with a JSON string with a key 'answer'. The "answer" should follow the format of the concise answer in the reasoning path. Unless the query is a description task, the "answer" should only consist of one or several words or numbers that indicates only one answer to the query directly.

Figure 15: Prompt for final reasoning and answer generation.

