# OpenReview forum: "ChemisTRAG: Table-based Retrieval-Augmented Generation for Chemistry Question Answering"
_ICLR.cc/2026/Conference — Submitted to ICLR 2026_

### Official Review · Reviewer_sht6 · 2025-10-14

**Soundness:** 4
**Presentation:** 3
**Contribution:** 4
**Rating:** 6
**Confidence:** 3

**Summary:**

This paper introduces ChemisTRAG, a table-based retrieval-augmented generation (RAG) system designed to improve large language model (LLM) performance on chemistry question answering. Unlike prior text-based RAG methods, ChemisTRAG structures chemical knowledge as relational tables of compounds and reactions, supporting schema-aligned retrieval and adaptive reasoning. The authors also construct an up-to-date patent-based chemistry knowledge base and an accompanying benchmark dataset covering eight QA tasks. Experimental results show substantial improvements over text-based RAG across multiple LLMs and input representations (IUPAC and SMILES), suggesting the proposed framework enhances both retrieval precision and reasoning robustness.

**Strengths:**

The work is well-motivated and addresses a genuine limitation of text-based retrieval in chemistry—semantic mismatches caused by inconsistent naming conventions. The proposed tabular paradigm is intuitive and demonstrates meaningful gains across benchmarks. The paper’s data construction pipeline is transparent and its systematic evaluation across tasks, input types, and models provides comprehensive evidence of performance improvements. The inclusion of case studies showing both exact-match and out-of-corpus reasoning illustrates the framework’s practical benefits.

**Weaknesses:**

Despite its novelty, the work lacks theoretical or methodological depth beyond the structural organization of data and prompt engineering. The adaptive reasoning component is largely heuristic, without clear algorithmic grounding or ablation beyond basic variants. The evaluation benchmark relies heavily on synthetic, LLM-generated questions rather than human-curated or real-world chemical problems, potentially inflating reported performance. The paper also omits a rigorous error analysis, making it difficult to assess when and why the system fails. Finally, the scope of the claimed improvement (“generalizable framework for chemistry”) seems overstated given that the experiments are limited to relatively simple QA templates rather than true open-domain reasoning.

**Questions:**

1. The “adaptive reasoner” lacks formal specification—its “prototype–grounding–execution” process reads more like prompt engineering than a distinct algorithm. Quantitative justification for each stage is limited.
2. The retrieval evaluation is strong, but no clear comparison is made with domain-specific symbolic or graph-based retrievers, which are standard in cheminformatics.
3. The dataset heavily depends on LLM-paraphrased questions, introducing potential bias toward models with similar linguistic priors. Some human validation or leakage analysis would strengthen credibility.
4. Baselines are incomplete: recent chemistry-specific RAG or tool-augmented systems (e.g., ChemCrow, ChemAgent) are not directly benchmarked, weakening the comparative claim.
5. The discussion of limitations should be more explicit, including issues like LLM hallucination in synthesis prediction and the inability to handle multi-step reactions or mechanistic reasoning.

---

> ### Author Response · Authors · 2025-11-23
> **Response to Reviewer sht6 (1/2)**
>
> Thank you for your constructive comments. We first respond to your questions point by point (1-5) and then to the weaknesses (6-7), as some of the weaknesses are corresponding to a question.
>
> ## 1. Scientific Support of Our Method
> We support the design of our reasoning method from two perspectives: cognitive science and recent algorithmic advances:
> 1. **Cognitive Alignment**: Our "Planing-Grounding-Execution" workflow mimics the System 2 thinking process (Kahneman, 2011) and Polya’s classic problem-solving framework (Understand -> Plan -> Execute). In chemistry, experts usually first formulate a high-level strategy (the Prototype) and then verify it with specific data (the Grounding).
> 2. **Algorithmic Foundation**: This design is inspired by state-of-the-art reasoning paradigms for chemistry problems such as StructChem (Ouyang, 2024) and ChemAgent (Tang, 2025). StructChem demonstrates the effectiveness of a **reasoning structure** to guide knowledge elicitation, and our “planning stage” imposing a logical structure before generation. ChemAgent highlights that chemical tasks **require decomposing problems into sub-tasks**. Our method adopts this insight and innovates by adapting it for RAG.
>
> Additionally, we clarify that we have conducted a **quantitative ablation study** for each module (**Figure 5 and lines 437-448**). Here we provide a tabular version for your review:
> | Method               | In Corpus | Out of Corpus |
> |----------------------|-----------|---------------|
> | ChemisTRAG           | **67.19**     | **34.58**     |
> | w/o Prototype        | 64.35     | 31.33         |
> | w/o Adaptation       | 63.25     | 30.19         |
> | w/o All              | 60.28     | 28.08         |
>
>
> ## 2. Different Retrievers
> Given that in RAG systems, domain-specific symbolic or graph-based retrievers require complex query pre-processing (e.g., language to graph) and knowledge base reconstruction, this falls outside our research scope. Thus, we focused on text-based retrieval methods. However, we addressed the need for **domain-specific comparison** by adding a **ChemBERTa** baseline (pre-trained on massive chemical data) for matching extracted query compounds against KB entries (Equation 2).
> | Retrieval Method   | Reaction | Compound | Overall |
> |----------|----------|----------|---------|
> | ChemBERTa | 48.38    | 74.74    | 58.27   |
> | ROUGE-L  | 57.07    | 84.06    | 67.19   |
>
> The results show that our simple ROUGE-L based method outperforms the domain-specific encoder. This is likely because chemical identifiers (SMILES/IUPAC) possess a rigorous string structure where lexical overlap directly correlates with structural identity. In contrast, dense embeddings (like ChemBERTa) may introduce noise in entity linking..
>
> ## 3. Human Verification on Benchmark
> We invite **two chemistry PhD students** for a manual verification on **240 samples**.
> Each annotator judges whether (1) the question is natural and aligned with typical human user query style, and (2) the answer correctly and completely addresses the question. An item is accepted only if both criteria are satisfied.
>
> The results show **high reliability (230/240=95.8% pass rate)** of the sampled data and **high agreement** (97.5% of cases receiving matching judgments) between the two experts.
>
> ## 4. More Baselines
> We compare ChemisTRAG with several recent baselines ChemRAG (external knowledge), ChemCrow (external tools), and ChemAgent (reasoning optimization):
> | Method | Reaction | Compound | Overall |
> | --- | ---: | ---: |   ---: |
> | Direct | 23.74 | 27.18 | 25.02 |
> | ChemCrow | 24.55 | 31.21 | 27.05 |
> | ChemAgent | 28.77 | 33.82 | 30.66 |
> | ChemRAG | 36.44 | 46.28 | 40.14 |
> | ChemisTRAG | **57.07** | **84.06** | **67.19** |
>
> As shown in the results, ChemisTRAG **outperforms all baselines**, demonstrating the effectiveness of our table-based retrieval and adaptive reasoning framework.
>
> ## 5. Discussions of Limitations
> Due to the space constraint, we haven’t conducted specific and in-depth discussions. We fully agree with the discussions you proposed and will incorporate them:
> - **LLM Hallucination**: While RAG grounds generation to mitigate hallucinations, intrinsic LLM flaws persist. Future work may implement hallucination detection or multi-model checks to further reduce this risk.
> - **Multi-step Reactions & Reasoning**: Our current work focuses on single-step QA. Addressing **multi-step reactions and mechanistic reasoning** is a crucial future challenge. This may require collecting expert-annotated **CoT** data and exploring methods like **Reinforcement Learning** to enhance long reasoning.

---

> > ### Author Response · Authors · 2025-11-23
> > **Response to Reviewer sht6 (2/2)**
> >
> > ## 6. Error Analysis
> > To provide a concrete view of system limitations, we analyzed typical failure cases. A primary error source is **"Reasoning under Noisy Context,"** where the retriever finds the correct entry but also introduces highly similar "distractors," causing the LLM to hallucinate or select the wrong row.
> >
> > **A Typical Failure Case (Distractor Interference)**
> >
> > | Query | Retrieved Evidence (Top-2) | Model Response (Error) |
> > | :--- | :--- | :--- |
> > | What's the molecular mass of CN1C=CC=C1? | [Target] CN1C=CC=C1 (Mass: 81.12) [Noise] CN1C=CC=N1 (Mass: 82.1) | "{reasoning: "...As retrieved, the mass of CN1C=CC=N1 is 82.1…", answer: **82.1**}" |
> >
> > In this case, although the correct compound was retrieved, the presence of a similar distractor (CN1C=CC=N1) confused the reasoner, leading to an incorrect answer. This highlights the need for finer-grained selection logic in future work.
> >
> > ## 7. Scope of Contribution
> > We agree that our work focuses on chemistry QA tasks, so our title, abstract, and main text emphasize this focus rather than a "generalizable framework for chemistry." As mentioned in point 5, we will discuss exploring the application of this framework to more complex chemistry tasks in future work.
> >
> > ## Conclusion
> > In summary, we attempt to address your concerns by adding manual validation, conducting additional experiments, providing more in-depth analysis, and clarifying certain issues. We welcome your further comments and suggestions, which are of great significance to our work.
> >
> > ## References
> > [1] Kahneman, Daniel. Thinking, fast and slow. macmillan, 2011.
> >
> > [2] Polya, George. "How to solve it." 1957.
> >
> > [3] Ouyang, Siru, et al. "Structured Chemistry Reasoning with Large Language Models." ICML, 2024.
> >
> > [4] Tang, Xiangru, et al. "ChemAgent: Self-updating Library in Large Language Models Improves Chemical Reasoning." ICLR 2025.

---

### Official Review · Reviewer_VY3c · 2025-11-01

**Soundness:** 2
**Presentation:** 2
**Contribution:** 2
**Rating:** 4
**Confidence:** 5

**Summary:**

The paper proposes ChemisTRAG, a structured retrieval-augmented generation framework for chemistry question answering. The authors argue that text-based RAG systems struggle in chemistry because molecule names and formats (IUPAC, SMILES, etc.) vary too much for reliable matching. Their fix is to store everything in tables instead of free text. The setup includes a knowledge base built from USPTO patents (boiled down to 38k compounds and 23k reactions), a retriever that turns natural-language questions into structured queries aligned with the table schema, and an “adaptive reasoner” that plans, grounds, and executes the final answer. They also release a benchmark of 4,800 QA pairs built from this same database. Experiments show large gains over text-based RAG, especially for SMILES inputs, and ablations suggest both the structured retriever and the reasoning module matter. Overall, it’s a cleanly engineered system showing that explicit tabular structure helps RAG work better in chemistry, though the setup feels quite controlled and domain-specific.

**Strengths:**

The paper’s main strength is its clear and well-engineered demonstration that structuring chemical knowledge into tables helps large language models handle the inconsistent naming and representation issues that often break text-based RAG systems. The design is neat and internally consistent — the knowledge base, retriever, and reasoner are all built to fit together, and the evaluation convincingly shows that this setup works much better than plain text retrieval when the data is well covered. The benchmark and data construction are also useful contributions on their own, giving a reproducible setup for chemistry QA. While the evaluation is somewhat closed-world, within that controlled scope the system is carefully implemented, the experiments are extensive, and the improvements over text RAG are large and consistent. Overall i think the paper’s strength lies in its solid engineering, clean framing of the structured retrieval problem and empirical validation in a chemistry setting.

**Weaknesses:**

The main weakness of the paper is that the benchmark and method are tightly coupled — the dataset is built directly from the same database the system retrieves from, so all entities are guaranteed to exist and appear in nearly identical form. This makes retrieval artificially easy and inflates performance, showing the system’s ability to match known strings rather than handle realistic, noisy, or missing information. The “adaptive reasoning” module also feels more like general prompt engineering than a novel algorithmic idea, and its contribution is not specific to chemistry or the tabular setup. Finally, the in-corpus vs. out-of-corpus experiment meant to show robustness under imperfect information is underdeveloped: correctness is ill-defined, the results barely differ from the non-RAG setup (table 2), and the section reads more as a late-added completeness check than as real evidence of reasoning with missing data.

**Questions:**

Since entities in the benchmark originate from your KB and are later paraphrased, how do you ensure that entity extraction and linking during evaluation are not trivially solved by lexical overlap? Was any normalization or perturbation (e.g., name variations, typos) introduced to test robustness?

How exactly do you determine that an answer is “correct”? Is correctness based on exact string match?

In Section 5.4, how do you operationally define “reasoning with imperfect information”? When the gold KB entry is removed, what does a correct response look like — a partial answer, an uncertainty statement, or a best guess? Could you clarify how you interpret near-identical performance between the two settings?

---

> ### Author Response · Authors · 2025-11-23
> **Response to Reviewer VY3c (1/2)**
>
> First, we respond to your questions point by point (1-3), as some of them correspond to certain weaknesses, and then we address the remaining weaknesses (4).
> ## 1. Reliability of Evaluation
> We structure our response to this concern in two parts: design rationale and efforts to ensure generalization of evaluation.
> ### Design Rationale
> The coupling of the knowledge base (KB) and the benchmark introduces some degree of lexical overlap, which can simplify entity linking. Our primary goal was to explore the contribution of **structured retrieval** to LLM performance. The controlled environment, where the correct knowledge exists and can be reliably linked, allowed us to perform a clear study of the RAG mechanism. However, we recognize that this may lead to the concern about the **generalizability of the evaluation**. Therefore, we conducted two experiments below to support the effectiveness of our method.
>
> ### Generalization and Robustness
> **Out-of-Corpus (OOC) Performance**: As shown in Figure 5, ChemisTRAG provides performance gains (34.58 v.s 25.02) even when the gold answer is not present in the knowledge base. This demonstrates that the system utilizes retrieved evidence for adaptive reasoning, rather than merely performing a string lookup.
>
> **Robustness against Perturbation (New Experiment)**: To directly address the concern regarding robustness on noise, we conducted an additional experiment, introducing **perturbations (typos and name variations)** to the chemical entities in a 15% random sample of the benchmark.Then, we evaluate our method on the perturbed data under the same settings.
>
> |        | Reaction       |        |        |        |        | Molecule       |        |        | Overall | Original (No Noise)
> |----------------|--------------------------|--------|--------|--------|--------|-------------------------|--------|--------|---------|---------|
> |  | PP    | RP    | CoP    | SP    | CaP    | WC     | NC     | MD     |         |         |
> | w/o RAG         | 19.48 | 32.62 | 23.33  | 17.02 | 13.12  | 24.44  | 23.49  | 24.88  | 22.30   | 25.02 |
> | w/ TextRAG     | 25.94 | 34.19 | 42.22  | 40.88 | 27.84  | 48.89  | 40.23  | 43.35  | 37.94   | 40.14|
> | w/ ChemisTRAG  | 46.14 | 48.34 | 53.33  | 65.11 | 53.74  | 86.67  | 78.51  | 74.67  | **63.31**   | **67.19** |
>
>
> While noise naturally attenuates performance across all methods (25->22, 40->37, 67->63), ChemisTRAG maintains a clear performance margin over the baselines (63 v.s 37).
>
> *Justification of “15%”: 1. BERT (Devlin et al., 2019) introduces 15% noise during training. 2. Some work (Pinet et al., 2022) finds that the rate of typo errors ranges between 12% and 20%. Thus, 15% should be a reasonable setting.*
>
> ## 2. Metrics
> As shown in lines 313-317 in our paper, the metric depends on the answer type: **String Answers** (e.g., natural language, IUPAC name): ROUGE-L; **Numerical Answers**: tolerance-based exactness (correct/incorrect within a small margin); **SMILES Answers**: molecular similarity computed by RDKit. Final results are presented as averages.
>
> ## 3. “Out of Corpus” Settings and Results
> We appreciate the request for clarification on our evaluation under partial knowledge.
> 1. **Operational Definition of Imperfect Information**: The term "reasoning with imperfect information" is operationalized by our **Out-of-Corpus (OOC) setting**. As described in line 435, this is achieved by **removing the exact KB entries** corresponding to the ground truth answer. This setting forces RAG systems to perform *inference using only partial, analogous, or contextually associated evidence remaining in the KB*.
> 2. **Correct OOC Response**: The ground truth and evaluation metrics in the OOC setting **remain unchanged** to the In-Corpus setting. Thus, under OOC, a correct response is still defined as the response that **matches the ground truth**.
> 3. **Clarification on OOC Performance**: We clarify the performance under the OOC setting. As shown in **Figure 5**, ChemisTRAG's performance (orange bar at the bottom) in the OOC setting (**34.58**) is **not** "near-identical" to the non-RAG baseline (Direct Reasoning, red dashed line: **25.02**). Here we provide a tabular result for your review:
> | Method               | In Corpus | Out of Corpus |
> |----------------------|-----------|---------------|
> | w/o RAG              | 25.02     | 25.02         |
> | w/ ChemisTRAG           | **67.19**     | **34.58**     |

---

> > ### Author Response · Authors · 2025-11-23
> > **Response to Reviewer VY3c (2/2)**
> >
> > ## 4. Novelty of Adaptive Reasoner
> > We clarify that our **Adaptive Reasoner** is not merely generic prompt engineering, but aligns with the **domain-specific consensus** established by recent SOTA chemistry reasoning frameworks.
> > - **StructChem**  (Ouyang et al., ICML 2024) emphasizes that to solve chemistry problems, LLMs require an explicit "reasoning structure" to construct logic generation.
> > - **ChemAgent** (Tang et al., ICLR 2025) proves that "task decomposition" is essential to prevent cascading errors in multi-step chemical reasoning.
> >
> > These studies inspired our adaptive reasoner. Differently, we **decouple** the generation of abstract chemical logic (e.g., reaction pathways, problem-solving steps, chemistry formulas) from the retrieval of scientific facts (e.g., reactants, conditions, specific SMILES, numerical properties). From a research perspective, we explore the **balance** between LLM's reasoning ability and the utilization of external knowledge, going beyond prompt engineering.
> >
> > In general, we hope our responses address your questions and resolve your concerns. We welcome your further comments, which are highly beneficial to our work.
> >
> > ## References
> > [1] Jacob, Devlin et al. “BERT: Pre-training of Deep Bidirectional Transformers for Language Understanding” NAACL 2019.
> >
> > [2] Svetlana, Pinet et al. “Typing expertise in a large student population.” Cognitive Research: Principles and Implications 2022.
> >
> > [3] Siru, Ouyang et al. "Structured Chemistry Reasoning with Large Language Models." ICML, 2024.
> >
> > [4] Xiangru, Tang et al. "ChemAgent: Self-updating Library in Large Language Models Improves Chemical Reasoning." ICLR 2025.

---

### Official Review · Reviewer_vdLk · 2025-11-03

**Soundness:** 3
**Presentation:** 3
**Contribution:** 3
**Rating:** 4
**Confidence:** 2

**Summary:**

The authors propose ChemisTRAG, a Retrieval-Augmented Generation (RAG) system designed to address the unique challenges of question answering in the chemistry domain1. The core problem, as identified by the authors, is that standard text-based RAG systems fail to align queries with knowledge corpora when chemical entities are represented in diverse and non-linguistic formats, such as IUPAC names and SMILES strings. To solve this, the proposed system has three main components: a Tabular Knowledge Base (KB), a Table-Based Retriever, and an Adaptive Reasoner. To evaluate the system, the authors also create a new benchmark dataset of 4,800 QA pairs derived from their KB, covering eight distinct task types.

**Strengths:**

- The paper targets a clear and important weakness of general-purpose RAG systems. The difficulty of aligning symbolic or non-standard identifiers (like SMILES) with text is a real-world bottleneck in many scientific and technical domains, and the authors' solution of using a structured, table-based KB is a highly pragmatic and effective approach.

- The authors contribute two valuable, up-to-date resources: the tabular KB of 38k+ compounds and 23k+ reactions from recent patents (2020-2025) and a new 4,800-pair benchmark dataset for chemistry QA. These resources are a solid contribution to the community, independent of the method itself.

**Weaknesses:**

- The components of ChemisTRAG are practical engineering, but they are not new methods.
- The entire data pipeline (KB extraction and benchmark generation) is automated using LLMs (Qwen-3-8B and GPT-OSS-20B). The "validation" step for the KB also uses an LLM. This introduces a significant risk of systemic error, hallucination, and self-reinforcing biases. The paper provides no statistics on human-in-the-loop verification or expert validation of the final KB and benchmark. Without this, the "ground truth" of the dataset is questionable.

**Questions:**

See weaknesses

---

> ### Author Response · Authors · 2025-11-23
> **Response to Reviewer vdLk**
>
> ## 1. Novelty of ChemisTRAG
> We clarify that ChemisTRAG provides methodological novelty by establishing a paradigm to solve intrinsic failures of generic RAG in chemistry:
> - **Tabular Retrieval (Solving the Terminology Bottleneck)**: Generic text-RAG struggles with the "many-to-one" nature of chemical names (e.g., SMILES vs. Synonyms). We propose Schema-Aligned Retrieval, which parses natural language into structured tuples. This is not just engineering, but a specific algorithmic solution to ensure precise knowledge matching in a domain where semantic vectors often fail.
> - **Adaptive Reasoning (Balancing Logic and Evidence)**: We address an underexplored issue: LLMs tend to over-rely on retrieved evidence, suppressing their own reasoning. Our "Planning-Grounding-Execution" reasoning decouples the process. Inspired by but distinct from StructChem’s structured reasoning (Ouyang et al., ICML 2024) and ChemAgent’s task decomposition (Tang et al., ICLR 2025), we require the LLM to formulate a logical skeleton before utilizing retrieved data, thereby effectively balancing its internal reasoning capabilities with external knowledge.
> - **Foundational Framework**: As a pilot study, ChemisTRAG demonstrates the superiority of structured over unstructured RAG in science, serving as a foundational baseline to steer future research towards higher precision.
>
> ## 2. Reliability of the Benchmark
> **Human Verification**: We appreciate this critical suggestion. We invited **two chemistry PhD students** to verify that the **240** randomly sampled QA pairs cover all 8 tasks. Each annotator independently judged whether (1) **Naturalness**: the question is natural and aligned with typical human user query style, and (2) **Correctness**: the answer correctly and completely addresses the question. An item is accepted only if both criteria are satisfied.
>
> This verification yields a **95.8% pass rate (230/240)** with 97.5% inter-annotator agreement. This high level of expert consensus shows that our automated construction pipeline produces reliable and chemically sound ground truth data.
>
> **Evaluation via External Benchmarks**: To further alleviate concerns about "self-reinforcing biases" (i.e., the model only working on its own generated data), we evaluated ChemisTRAG on **public, external benchmarks** that were not created by our pipeline: **ChemBench** and **SciBench**.
>
> | Method | ChemBench | SciBench |
> |----------------|-----------|----------|
> | w/o RAG | 55.38 | 30.57 |
> | w/ ChemRAG | 57.63 | 32.75 |
> | w/ ChemisTRAG (Ours) | **60.25** | **34.93** |
>
> ChemisTRAG outperformed the ChemRAG baseline on both ChemBench (**60.25** vs 57.63) and SciBench (**34.93** vs 32.75). Since our system performs well on external, human-curated datasets, it proves that our method is robust and not overfitting to artifacts in our generated training data.
>
> We hope the combination of **expert human verification** and **strong performance on external public benchmarks** addresses the concerns regarding the reliability and generalization of our work.
>
> ## References
> [1] Ouyang, Siru, et al. "Structured Chemistry Reasoning with Large Language Models." ICML, 2024.
>
> [2] Tang, Xiangru, et al. "ChemAgent: Self-updating Library in Large Language Models Improves Chemical Reasoning." ICLR 2025.

---

### Official Review · Reviewer_bf2g · 2025-11-04

**Soundness:** 2
**Presentation:** 3
**Contribution:** 2
**Rating:** 4
**Confidence:** 4

**Summary:**

This paper presents ChemisTRAG, a table-based retrieval-augmented generation (RAG) framework designed for chemical reaction and compound question answering. Instead of relying on unstructured text retrieval, the authors construct a structured knowledge base containing ~23K curated reaction records and ~38K unique compounds extracted from recent patent and literature sources. Natural language queries are parsed into structured tuples that specify entity type and query intent, allowing retrieval to be performed over relational tables rather than documents. Retrieved evidence is then combined with an adaptive reasoning module to generate the final answer. Experiments show that ChemisTRAG achieves higher accuracy and robustness than text-based RAG baselines and domain LLMs. However, the experiments are limited to the dataset generated in this paper, without evaluation on other chemistry datasets. The generated evaluation dataset is constructed from the knowledge base in this paper, raising concerns about generalization. The generated dataset also misses some critical chemistry tasks, including yield prediction, chemistry calculation, and general academic chemistry problems.

**Strengths:**

- The authors propose a table-based retriever for retriever from the constructed knowledge base, and also take synonym problem into consideration
- The experiments on the retrieval are comprehensive
- The paper is well written and easy to follow

**Weaknesses:**

- The evaluation data is constructed from the KB in this paper, raising concerns about generalization.
- The proposed method is only evaluated on the data created in this paper. There are lots of public chemistry datasets[2][3][4][5], but the authors choose not to use any of them. This decreases the credibility of this paper.
- The proposed adaptive reasoner is not very novel, many agents use similar strategies, e.g. ChemAgent [1].
- For compound-focused queries, the proposed method simply uses ROUGE-L similarity. This may not be good enough since there are lots of chemistry specific encoders, including ChemBERTa, MolT5, and MolBERT. These encoders may provide better performance.
- Given how the knowledge base is constructed, I worry that the generalization is not so good. The proposed evaluation dataset misses some critical tasks existed in other works, including yield prediction, general college chemistry questions, and chemistry calculation problems.


[1] Tang et al. ChemAgent: Self-updating Library in Large Language Models Improves Chemical Reasoning. ICLR 2025.

[2] Wang et al. SciBench: Evaluating College-Level Scientific Problem-Solving Abilities of Large Language Models. ICML 2024.

[3] Fang et al. Mol-Instructions: A Large-Scale Biomolecular Instruction Dataset for Large Language Models. ICLR 2024.

[4] Mirza et al. Are large language models superhuman chemists? Nature Chemistry 17, 984–985 (2025).

[5] Rein et al. GPQA: A Graduate-Level Google-Proof Q&A Benchmark. COLM 2024.

**Questions:**

- How do you check the chemical feasibility of extracted reaction with GPT-OSS-20B? What are you validating here? Wouldn't using Molecular Transformer more reasonable here?
- What retriever do you use for TextRAG?
- What are the formats of the constructed questions? Are they multiple-choice questions? If not, what metrics do you use for each task?

---

> ### Author Response · Authors · 2025-11-23
> **Response to Reviewer bf2g (1/2)**
>
> # To Weaknesses
>
> ## 1.Generalization of Evaluation
> We acknowledge the reviewer's concern regarding generalization due to the KB-sourced nature of our benchmark. We provide strong empirical evidence that our system's benefits extend beyond trivial string matching:
> 1. **Out-of-Corpus (OOC) Evaluation (Figure 5)**: ChemisTRAG demonstrates significant performance gains over the baseline without RAG (25.02 -> 34.58) even when the required **ground truth entity is removed from the KB**. This demonstrates the ChemisTRAG's ability to perform analogous reasoning based on partial evidence. Here we present a tabular result for your review:
>
> | Method               | In Corpus | Out of Corpus |
> |----------------------|-----------|---------------|
> | ChemisTRAG           | **67.19**     | **34.58**     |
> | w/o Prototype        | 64.35     | 31.33         |
> | w/o Adaptation       | 63.25     | 30.19         |
> | w/o All              | 60.28     | 28.08         |
>
> 2. **Robustness Against Perturbation (New Experiment)**: To directly address input noise, we conducted an additional experiment by introducing **random typos and name variations** to chemical entities in a 15\% sample. We tested our method on this perturbed data under the same setup. As shown in the new results, ChemisTRAG maintains a clear performance margin over baselines despite the input variance, demonstrating **robustness** against real-world entity variations.
> **Results on Evaluation Data with Perturbation:**
>
> |        | Reaction       |        |        |        |        | Molecule       |        |        | Overall | Original (No Noise)
> |----------------|--------------------------|--------|--------|--------|--------|-------------------------|--------|--------|---------|---------|
> |  | PP    | RP    | CoP    | SP    | CaP    | WC     | NC     | MD     |         |         |
> | w/o RAG         | 19.48 | 32.62 | 23.33  | 17.02 | 13.12  | 24.44  | 23.49  | 24.88  | 22.30   | 25.02 |
> | w/ TextRAG     | 25.94 | 34.19 | 42.22  | 40.88 | 27.84  | 48.89  | 40.23  | 43.35  | 37.94   | 40.14|
> | w/ ChemisTRAG  | 46.14 | 48.34 | 53.33  | 65.11 | 53.74  | 86.67  | 78.51  | 74.67  | **63.31**   | **67.19** |
>
>
> *Justification of “15%”: 1. BERT (Devlin et al., 2019) injects 15% noise during training. 2. Some work (Pinet et al., 2022) finds that the rate of typos ranges between 12% and 20%. Therefore, “15%” should be a reasonable setting.*
>
> ## 2. Performance on Other Benchmarks
> We agree that evaluating on public datasets is crucial for credibility. To address this, we conducted additional experiments on two benchmarks used in ChemRAG (Zhong et al., 2025), **ChemBench** (Zhang et al., 2024) and **SciBench-Chemistry** (Wang et al., 2024), covering tasks from reaction prediction (including yield prediction) to college-level chemistry problems. We utilized the same backbone model (Qwen) and compared our method against a standard text-based RAG baseline (**ChemRAG**).
>
> | Method | ChemBench | SciBench |
> |----------------|-----------|----------|
> | w/o RAG | 55.38 | 30.57 |
> | w/ ChemRAG | 57.63 | 32.75 |
> | w/ ChemisTRAG (Ours) | **60.25** | **34.93** |
>
> **ChemisTRAG outperforms both the non-RAG baseline and the text-based ChemRAG**. This shows that our **structured retrieval paradigm** generalizes well to external datasets, providing robust benefits for diverse chemical tasks beyond our constructed data.
>
> ## 3. Novelty of Adaptive Reasoner
> Our method draws inspiration from the **structured reasoning of StructChem** (Ouyang et al., 2024) and the **task decomposition of ChemAgent** (Tang et al., 2025). However, our focus and implementation differ significantly:
> - **Different Objectives**: While StructChem and ChemAgent focus on LLM **self-improvement** via internal reflection or tool use, ChemisTRAG addresses a unique challenge in RAG: **balancing internal reasoning with external evidence** (Lines 272-277).
> - **Novel Design**: Our core contribution is **decoupling logic generation from knowledge injection**. By forcing the LLM to first generate a "reasoning skeleton" (using internal knowledge) before populating it with retrieved data (external knowledge), we prevent the common RAG failure mode where models become over-reliant on retrieved context and lose their multi-step reasoning capabilities. This specific mechanism for **Reasoning-Retrieval balancing** is distinct from the self-correction mechanisms in ChemAgent.

---

> ### Author Response · Authors · 2025-11-23
> **Response to Reviewer bf2g (2/2)**
>
> ## 4. Domain-Specific Encoder for Retrieval
> To address this, we compared our method with **ChemBERTa** (domain-specific dense retriever). The results reveal an interesting trade-off:
>
> | Setting       | Method    | Reaction | Compound | Overall |
> |---------------|----------|----------|----------|---------|
> | In-Corpus     | ChemBERTa| 48.38    | 74.74    | 58.27   |
> | -    | ROUGE-L  | **57.07**    | **84.06**    | **67.19**   |
> | Out-of-Corpus | ChemBERTa| **39.72**    | 33.20    | **37.27**   |
> |- | ROUGE-L  | 30.22    | **41.85**    | 34.58   |
>
> **Finding**: While ChemBERTa is superior for finding semantic relevance in unknown reactions (OOC), ROUGE-L surprisingly outperforms embeddings for compound retrieval in both settings.
>
> **Analysis**: Lexical overlap in SMILES (e.g., CCC vs CCCC) usually serves as a strong proxy for structural similarity in retrieval tasks like weight computation, whereas semantic embeddings may retrieve structurally dissimilar but "conceptually related" molecules, which is less helpful for specific property reasoning.
>
> ## 5. Other Tasks
> We omitted certain tasks (e.g., yield prediction, college chemistry) because they are difficult to generate and validate. For example, yields in patent texts are often not explicitly provided but require complex calculations. Lines 620-623 presents this consideration. We prioritized an easily verifiable benchmark that is constructed by an entity extraction paradigm. Additionally, our benchmark includes a fundamental calculation task (weight computation).
>
> However, to demonstrate generalization across these tasks, we conducted new evaluations on public benchmarks: **ChemBench** and **SciBench** (Point 2).These datasets specifically cover **yield prediction, complex calculations, and college chemistry questions**. Our experiments show ChemisTRAG outperforms text-based baselines on both ChemBench and SciBench, empirically confirming that our method effectively generalizes to and improves performance on the wide scope of chemical problems.
>
> # To Questions
> 1. We use GPT-OSS-20B to validate extracted reaction entries by checking whether they are consistent with the original patent text. Samples that fail validation will be excluded. This task, grounded in the patents, is fundamentally an NLP problem that requires **strong language understanding** rather than chemistry prediction. Therefore, a general LLM is more suitable than domain-specific models such as Molecular Transformer.
>
> Furthermore, to ensure the chemical validity of QA pairs beyond the automated checks, we conducted a **human evaluation** with two chemistry PhD students. The experts manually verified 240 sampled pairs and confirmed a **95.8% accuracy rate** regarding chemical correctness and feasibility.
>
> 2. We use **Qwen3-Embedding (line 323)**, a SOTA embedding model, as the encoder for knowledge retrieval in TextRAG (ChemRAG).
>
> 3. We use **open-ended QA**: each question has a single correct answer but no predefined options. As described **lines 313-317**, evaluation depends on the answer type: **String answers**: ROUGE-L; **Numerical answers**: tolerance-based exactness (correct/incorrect within a small margin); **SMILES answers**: RDKit molecular similarity.
>
> In summary, we hope our response with clarifications and additional experiments addresses your concerns and we look forward to your further comments.
>
> ## References
> [1] Jacob, Devlin et al. “BERT: Pre-training of Deep Bidirectional Transformers for Language Understanding” NAACL 2019.
>
> [2] Svetlana, Pinet et al. “Typing expertise in a large student population.” Cognitive Research: Principles and Implications 2022.
>
> [3] Xianrui, Zhong et al. “Benchmarking Retrieval-Augmented Generation for Chemistry” COLM 2025.
>
> [4] Di, Zhang et al. “Chemllm: A chemical large language model”, 2024.
>
> [5] Xiaoxuan, Wang et al. “SciBench: Evaluating College-Level Scientific Problem-Solving Abilities of Large Language Models”. ICML 2024.
>
> [6] Siru, Ouyang et al. "Structured Chemistry Reasoning with Large Language Models." ICML 2024.
>
> [7] Xiangru, Tang et al. "ChemAgent: Self-updating Library in Large Language Models Improves Chemical Reasoning." ICLR 2025.

---

### Author Response · Authors · 2025-11-27
**Summary of Responses**

We summarize our responses to facilitate the discussion. We have made the revised paper, where **blue** highlights indicate new content, while **orange** highlights mark original text relevant to some questions.

## Reliability of Our Evaluation
### Quality of Our Benchmark (vdLk, sht6)
We conducted a **human evaluation** with **two chemistry PhD students** on 240 samples.
Each annotator independently judged whether (1) **Naturalness**: the question is natural and aligned with typical human user query style, and (2) **Correctness**: the answer correctly and completely addresses the question. An item is accepted only if both criteria are satisfied.
The results show **high reliability (95.8% pass rate)** of the sampled data and **high agreement** (97.5% of cases receiving matching judgments) between the two experts.

### Generalization of Evaluation (bf2g, VY3c)
Two reviewers expressed concerns about the coupling of the knowledge base and the evaluation set, arguing that it undermines the generalizability of the evaluation. To address this concern, we 1) evaluate our method on Out-of-Corpus (OOC) setting and 2) conduct an **additional evaluation** where we introduce **perturbation** (typos, name variants) into queries. Both experiments show that our method brings significant improvement to baselines, demonstrating the **effectiveness and robustness** of our method in realistic scenarios.

## Novelty
- **Reviewer vdLk** questions the novelty of our framework **ChemisTRAG**. We clarify that ChemisTRAG provides methodological novelty by **establishing a table-based** paradigm to solve intrinsic failures of RAG systems in chemistry.
- **Reviewers bf2g and VY3c** argue that our **reasoning method** is engineering-oriented and similar to previous work. We compare our work to StructChem and ChemAgent to position our reasoning method. We clarify that our innovation lies in the reasoning paradigm: by **decoupling reasoning** and **knowledge injection**, we achieve a balance between the utilization of internal LLM knowledge and external knowledge. This paradigm and research focus have not been explored before.
- **Reviewer sht6** acknowledges the **novelty** of our method but argues that it lacks **theoretical or algorithmic support**. To address this concern, we justify the design of our method from cognitive science and recent algorithmic advances.

## Variants of Retrievers (bf2g, sht6)
We mainly address the concerns about retriever variants and selection by incorporating **ChemBERTa** into the retrieval module. Experimental results demonstrate the effectiveness of our current method (ROUGE-L), as well as the generalization advantages of ChemBERTa as a chemistry-specific encoder in certain scenarios.

## More Benchmarks (bf2g)
To address the concern about the generalizability of our method, we evaluated our method on two widely-used benchmarks **ChemBench** and **SciBench**, which include some tasks that we haven’t included. Experimental results show the effectiveness of our method on these two datasets, demonstrating its generalizability.

## More Baselines (sht6)
We compare ChemisTRAG with several baselines such as **ChemRAG** (external knowledge), **ChemCrow** (external tools), and **ChemAgent** (reasoning optimization). Results show that ChemisTRAG outperforms all baselines, demonstrating the effectiveness of our table-based retrieval and adaptive reasoning framework.

## Clarification
We clarify that our paper has presented the details that address the concerns (highlighted with the orange color):
- **Metrics (bf2g, VY3c)**: Lines 339-342.
- **Comparison between In/Out of Corpus (VY3c)**: Figure 5 and Lines 441-442.
- **Ablation Study (sht6)**: Figure 5 and lines 427-431.

In summary, we address the reviewers’ concerns and questions by supplementing manual verification, additional experiments, as well as clarifying our design considerations and paper details. We appreciate the reviewers’ further comments and suggestions.

---

### Author Response · Authors · 2025-11-30
**Summary of Key Revisions, Contributions, and Clarifications**

Dear Area Chairs, Senior Area Chairs, and Program Chairs,

As the author-reviewer discussion has **prematurely ended**, preventing updates from reviewers, we provide this summary to alleviate your workload and facilitate your final assessment. We believe the improvements and additional experiments listed below **directly fulfill the specific requests raised by the reviewers** and effectively resolve the primary concerns:

* **Resolved Data Reliability Concerns (Addressing vdLk, sht6):** We conducted a **human evaluation** with chemistry PhD experts. The results confirmed a **95.8% pass rate** for data correctness and naturalness, with **97.5% inter-annotator agreement**, strictly validating the quality of our benchmark.

* **Proven Generalization on Public Benchmarks (Addressing bf2g, VY3c):** To address concerns about generalizability, we extended our evaluation to external public benchmarks (**ChemBench & SciBench**). ChemisTRAG consistently outperformed baselines on these datasets, proving its robustness beyond our internal data.

* **Established Novelty & Specificity:** We clarified that our **"Tabular Paradigm"** and **"Adaptive Reasoning"** are methodological innovations specifically designed to solve the semantic mismatch and reasoning-retrieval imbalance unique to chemical RAG.

* **Clarifications on Evaluation Details:** We resolved technical **misunderstandings** regarding Metrics, In/Out-of-Corpus analysis, and Ablation Study by pointing to specific details already present in the original submission.

* **Value to the Community:** Beyond the method, we provide a Structured Knowledge Base covering **recent patents (2020-2025)**, an update over prevalent datasets often restricted to ~2016, addressing the scarcity of fresh resources in AI4Chemistry.

We believe these rigorous additions, **strictly following the reviewers' suggestions**, effectively address the key weaknesses and significantly strengthen the submission. **As reviewers were unable to update their assessments due to the premature conclusion**, the current scores do not reflect the strengthened submission. We trust your judgment in weighing these resolved concerns and the paper’s contribution to the community.

Best regards,

The Authors

---

### Meta-Review · Area_Chair_5Dpd · 2025-12-23

**Summary:**

This submission proposes ChemisTRAG, a table-based RAG framework for chemistry QA that stores compounds and reactions in a structured KB, retrieves schema-aligned rows, and uses an adaptive reasoning pipeline.

Reviewers agreed the paper is well-motivated and the engineering is clean, with strong empirical gains over text-RAG in the authors’ controlled benchmark.

The decision is driven by recurring concerns about **limited methodological novelty** (especially the “adaptive reasoner” reading as prompt/agent design), and about **evaluation credibility and generalization** because the benchmark is tightly coupled to an LLM-constructed KB and largely LLM-generated questions.

Reviewers also noted **missing or underdeveloped coverage** of broader chemistry tasks and limited evidence that the approach transfers beyond the curated setting.

**Reviewer Concerns:**

**Addressed in the rebuttal (partially to largely):**

* Reliability concerns about the automated pipeline were mitigated by a small-scale expert human check (240 samples) and added error analysis.
* Generalization concerns were partially addressed via (i) an out-of-corpus ablation, (ii) a perturbation study with typos/name variants, and (iii) added results on external benchmarks (ChemBench, SciBench), plus additional baselines and a ChemBERTa retriever comparison.

**Still outstanding:**

* The core evaluation remains substantially closed-world: the benchmark is derived from the same KB the model retrieves from, so entity linking and retrieval may be easier than in realistic chemistry QA where entities are missing, ambiguous, or require deeper normalization and chemical reasoning.
* The evidence for novelty remains limited. The table-centric storage and schema-aligned retrieval are practical, but reviewers were not fully convinced they constitute a new method beyond structured engineering choices; similarly, the adaptive reasoning pipeline remains close to established agentic prompting patterns without clear algorithmic or theoretical contribution.
* External benchmark gains are positive but modest, and it is unclear whether the proposed system would consistently outperform strong chemistry-specific retrieval and tool-based systems under more open-ended tasks (multi-step reasoning, mechanistic questions, calculations), which remain lightly validated here.
* The KB construction and validation still rely heavily on LLMs; the added human verification is helpful but may be insufficient to establish dataset and KB correctness at the scale claimed.

**Reviewer Scores:**

* **bf2g (rating 4, confident):** Likely increases slightly given the added external benchmark results, stronger baseline set, and ChemBERTa comparison. However, novelty and evaluation coupling concerns remain, so I expect maybe a move to 6, and plausibly staying at 4.
* **vdLk (rating 4, lower confidence):** The new human evaluation directly addresses their main concern about LLM-only validation, and the added public benchmark results help. I expect a modest upward shift, likely to 6.
* **VY3c (rating 4, very confident):** The perturbation experiment answers the robustness question, and the clarification of metrics helps. Still, their strongest critique is the tight KB-benchmark coupling and limited evidence for “reasoning with imperfect information,” which is only partially resolved. I expect little change, likely staying at 4.
* **sht6 (rating 6, moderate confidence):** The added baselines, human verification, and error analysis align with their requests and strengthen the submission. Their novelty and depth concerns persist, so I would expect them to remain around 6.

---

### Decision · Program_Chairs · 2026-01-26

Reject